# The mammalian rod synaptic ribbon is essential for Ca$_v$ channel facilitation and ultrafast synaptic vesicle fusion

**Chad Paul Grabner[1,2,3]\*, Tobias Moser[1,2,3,4]**

[1]Institute for Auditory Neuroscience and InnerEarLab, University Medical Center Göttingen, Göttingen, Germany; [2]Synaptic Nanophysiology Group, Max Planck Institute of Biophysical Chemistry, Göttingen, Germany; [3]Collaborative Research Center 1286 'Quantitative Synaptology', University of Göttingen, Göttingen, Germany; [4]Multiscale Bioimaging Cluster of Excellence (MBExC), University of Göttingen, Göttingen, Germany

**Abstract** Rod photoreceptors (PRs) use ribbon synapses to transmit visual information. To signal 'no light detected' they release glutamate continually to activate post-synaptic receptors. When light is detected glutamate release pauses. How a rod's individual ribbon enables this process was studied here by recording evoked changes in whole-cell membrane capacitance from wild-type and ribbonless (*Ribeye*-ko) mice. Wild-type rods filled with high (10 mM) or low (0.5 mM) concentrations of the Ca$^{2+}$-buffer EGTA created a readily releasable pool (RRP) of 87 synaptic vesicles (SVs) that emptied as a single kinetic phase with a $\tau$ <0.4 ms. The lower concentration of EGTA accelerated Ca$_v$ channel opening and facilitated release kinetics. In contrast, ribbonless rods created a much smaller RRP of 22 SVs, and they lacked Ca$_v$ channel facilitation; however, Ca$^{2+}$ channel-release coupling remained tight. These release deficits caused a sharp attenuation of rod-driven scotopic light responses. We conclude that the synaptic ribbon facilitates Ca$^{2+}$-influx and establishes a large RRP of SVs.

**\*For correspondence:** chadgrabner@gmail.com

**Competing interest:** The authors declare that no competing interests exist.

## Introduction

Animals use their sensory systems to interact with and navigate through their environment, and sensory maps are created for this purpose. This is especially true for vision, where perception of a real-world scene invariably asserts the location of objects in space. The building blocks for visual percepts originate from a visual field that is often in motion, and can vary greatly in luminance. Therefore, vertebrates have evolved complex processes that stabilize the eyes on the visual field (*Straka and Baker, 2013*), focus images on the back of the eye, and transform light of varying intensities into neural signals (*Rivlin-Etzion et al., 2018*).

The mammalian neural retina lines the concaved inner surface of the eye, and it forms a thin, multi-layered network. The outermost layer is a dense lawn of photoreceptors (PRs) (PR> $1\times10^5$ mm$^{-2}$). Each PR has a photosensitive outer segment, and at its opposite pole a single synaptic terminal forms in the first synaptic layer of the retina, the outer plexiform layer (OPL). There are two classes of PRs in the outer retina: rods and cones, which differ in several ways (for review, *Grünert and Martin, 2020*). For example, rods are ~$10^3$-fold more sensitive to light than cones (*Cao et al., 2014*), they only express rhodopsin while cones express one of the multiple types of opsin (*Fain et al., 2010*), and they outnumber cones at a ratio of 30:1 (*Grünert and Martin, 2020*). These features allow the outer retina to begin sorting light properties into neural signals. A variety of bipolar cell types carry PR signals to the inner retina (*Light et al., 2012*; *Behrens et al., 2016*; *Tsukamoto and Omi, 2016*), whereupon a

plethora of chemical and electrical synapses are assembled into circuits (*Demb and Singer, 2015*) that serve in visual behaviors like night vision, color perception, and motion detection (*Sterling, 2013*).

PRs and bipolar cells form 'ribbon synapses' that are named after the electron-dense plate that projects from the presynaptic AZ into the cytoplasm. A subset of vertebrate sensory neurons express the protein ribeye, which is localized to ribbons (reviews: *Lagnado and Schmitz, 2015*; *Moser et al., 2020*). Deletion of the *Ribeye* gene eliminates synaptic ribbons (*Maxeiner et al., 2016*), but a unifying role for ribbons in synaptic transmission has not been identified. For instance, paired recordings between ribbonless rod bipolar cells (rbcs) and AII amacrine cells showed that synaptic transmission was greatly reduced without altering $Ca^{2+}$ currents. The release deficit was reasoned to result from an uncoupling of $Ca_v$ channels from SVs (*Maxeiner et al., 2016*). In contrast, ribbonless hair cells showed a milder impairment in exocytosis (*Becker et al., 2018*; *Jean et al., 2018*), and they produced well-defined substitute AZs that were largely capable of compensating for the loss of ribbon AZs (*Jean et al., 2018*).

More recent investigations into Ribeye-ko mice have used functional assays to probe how the ribbonless retinal circuitry behaves. First, recordings from on-α-ganglion cells showed that on-responses to increments in light were robust in the absence of ribbons; however, significant alterations were also documented (*Okawa et al., 2019*). Interpreting the results was challenged by the complexity of the retinal circuitry; in particular the overlap in rod and cone pathways (for review, *Seilheimer et al., 2020*). In a subsequent study based on electroretinograms *Fairless et al., 2020* assigned the deficits in ribbonless circuitry to the rod pathway; however, the defects in the pathway were not identified. Therefore, in the current study, we examined how the ribbon influenced transmitter release from mouse PRs, which has not been tested directly; and in addition, studied the biophysics of exocytosis from mammalian PRs about which relatively little is known.

Mammalian rods express a single, large horseshoe-shaped ribbon that surrounds one or two rod bipolar dendrites that are on average ~250 nm away from the ribbon (for mouse; *Hagiwara et al., 2018*). In the dark, rods are maximally depolarized to produce a steady influx of $Ca^{2+}$ that drives the continual turnover of SVs. This keeps synaptic glutamate high enough to activate the postsynaptic inhibitory mGluR6→TRPM1 pathway in rbc dendrites (*Koike et al., 2010*), which equates to the 'dark signal.' A weak flux of photons is sufficient to hyperpolarize the rod and momentarily slow exocytosis to create a 'light signal' (for review, *Field and Sampath, 2017*). Mathematical models have predicted that a rod ribbon needs to achieve a release rate ≥40 SVs-s⁻¹ for the rbc mGluR6 pathway to activate and create a dark signal (*Rao-Mirotznik et al., 1998*; *Hasegawa et al., 2006*).

In the current study, high-resolution measurements of evoked SV exocytosis were made directly from mouse rods. The results demonstrated that the mouse rod ribbon creates multiple, uniformly primed sites for the release of 87 SVs. Their $Ca_v1.4$ channels activated rapidly (~200 µs) and exhibited unique forms of facilitation that influenced ultrafast release. These features were dependent on the ribbon, as ribbonless rods formed a much smaller readily releasable pool (RRP), and lacked $Ca_v$ channel facilitation. The study provides experimental results that support longstanding proposals on the function of the rod ribbon synapse, and we discuss how synaptic ribbons contribute to retinal signaling.

## Results
### Super-resolution readout of SV turnover at an individual rod ribbon

The majority of rod somata reside in the outer nuclear layer (ONL) and send a spindly axon to their singular presynaptic terminal in the OPL, which contains an individual synaptic ribbon (*Figure 1A and B*). The minority of rod somata that lack an axon are positioned in the OPL, and they contain the synaptic ribbon within the soma compartment (*Figure 1A*). These axonless rods have been described before at the EM level (*Li et al., 2016*). We have previously reported making whole-cell, voltage-clamp recordings of $Ca^{2+}$-currents from them (referred to as the rod 'soma-ribbon' configuration) in an attempt to better control the membrane potential about the ribbon AZ (*Hagiwara et al., 2018*). This point is illustrated here by assessing the membrane time constant for the soma-ribbon configuration. A −10 mV voltage step from a holding potential of −70 mV relative to the $AgCl_2$ reference electrode, generated a rapidly decaying membrane current with a $\tau$ <30 µs (*Figure 1C*). Furthermore, the RC

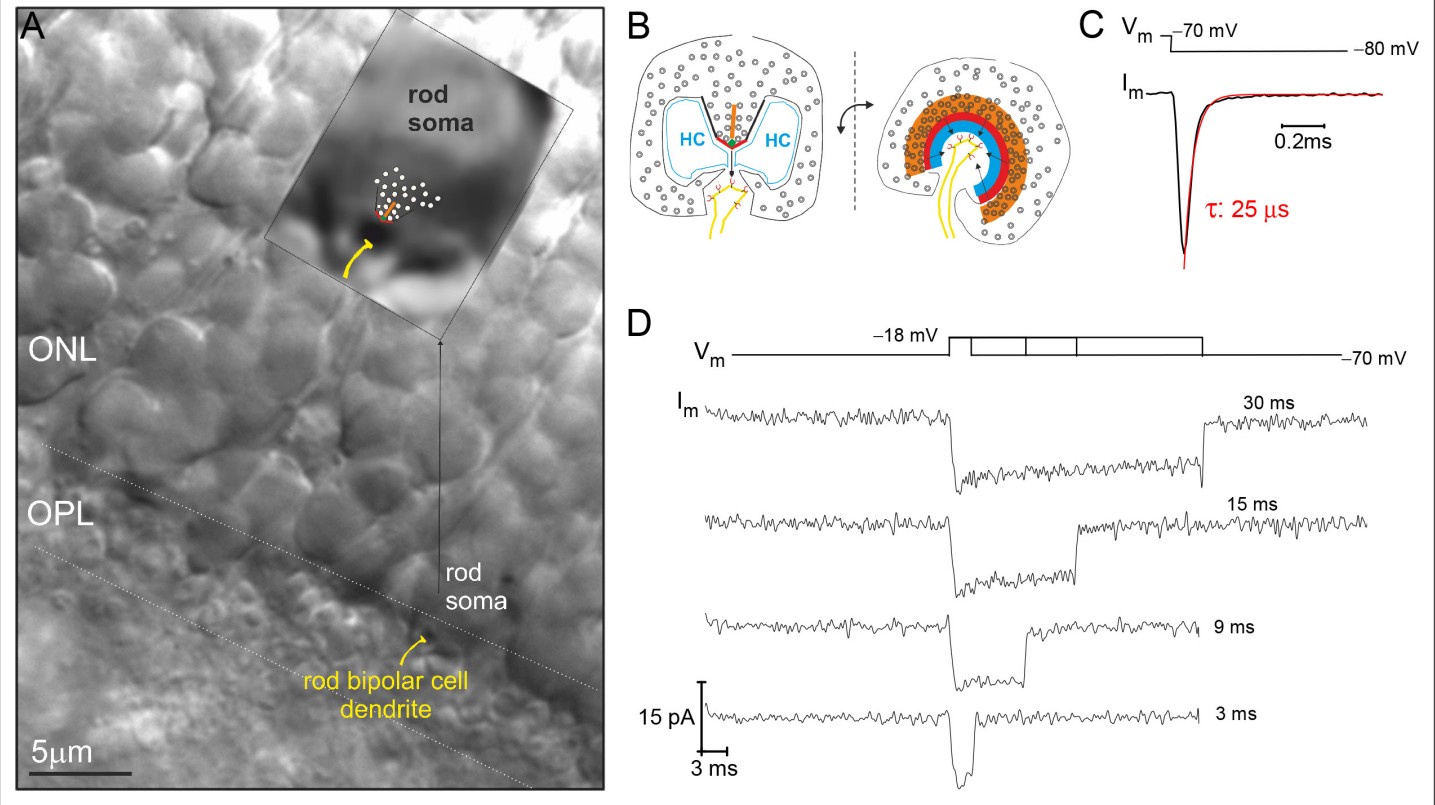

**Figure 1.** Example recording from a rod photoreceptor (PR) that lacks an axon. (**A**) Image of a retinal slice centered on PR terminals in the outer plexiform layer (OPL) and somata in the outer nuclear layer (ONL). The inset shows a zoomed in view of an axonless rod soma-ribbon in the OPL, to which the synaptic ribbon synapse has been drawn into the image for reference. (**B**) Illustration of the major components of the rod ribbon synapse. The two images are rotated by 90° relative to the plane of the ribbon. Legend: ribbon (orange), active zone (thick red line), arciform density (green diamond), ribbon flanked by synaptic ridges (thick black lines), horizontal cells (HC, in blue), and rod bipolar cell dendrite (yellow) with mGluR6 receptors (red). (**C**) The membrane current ($I_m$) transient measured from a rod soma-ribbon in response to a brief hyperpolarizing voltage step. Current trace taken prior to compensating whole-cell membrane capacitance ($C_m$). (**D**) Series of $Ca^{2+}$-currents measured from an individual rod in response to voltage steps for the indicated durations. See **Figure 2** for the corresponding evoked release from this rod.

time constant calculated from the average access resistance ($R_a$ ~30 MΩ) and whole-cell membrane capacitance ($C_m$ ~1 pF; **Supplementary file 1**) produces a $\tau_{RC}$ ~ 30 µs.

Being able to control and monitor membrane voltage with high temporal precision makes it possible to quantitatively study voltage-dependent $Ca^{2+}$-channels ($Ca_v$) and $Ca^{2+}$-triggered fusion of SVs. A series of voltage-dependent $Ca^{2+}$-current ($I_{Ca}$) traces, generated with depolarizing voltage steps ($V_{step}$) to −18 mV for different durations, highlights the tight voltage control of $I_{Ca}$ (**Figure 1D**; 10 mM EGTA in the whole-cell pipette). Next, to assess the extent of $Ca^{2+}$-triggered SV fusion associated with each depolarization, the sine wave-based, lock-in amplifier method was implemented to measure changes in whole-cell $C_m$ (**Lindau and Neher, 1988**). A depolarization evoked increase in $C_m$ can be attributed to the incorporation of SV membrane, as long as there are no corresponding changes in conductance that can interfere with the estimation of $C_m$ (see Materials and methods for details). The examples in **Figure 2A and B** illustrate these points. First, the response to a 9 ms step depolarization generated a robust increase in $C_m$, which remained elevated at a fixed level for the 0.5 s post-stimulation period captured in **Figure 2A**. Second, a sequence of evoked responses presented over a longer time span showed a pronounced jump in $C_m$ following each stimulation, while $G_m$ and $G_s$ were not noticeably changed by the stimulations (**Figure 2B**). This presentation also shows that subsequent to the evoked $\Delta C_m$, the membrane was endocytosed in the following ways: linearly, exponentially, and/or as abrupt downward steps (**Figure 2B**). Rarely were large (>2 fF) downward steps in $C_m$ observed, such as the one between the 3 and 9 ms stimulations in **Figure 2B** (downward arrow). Given the diversity of endocytotic responses, they were not quantified further in the current study.

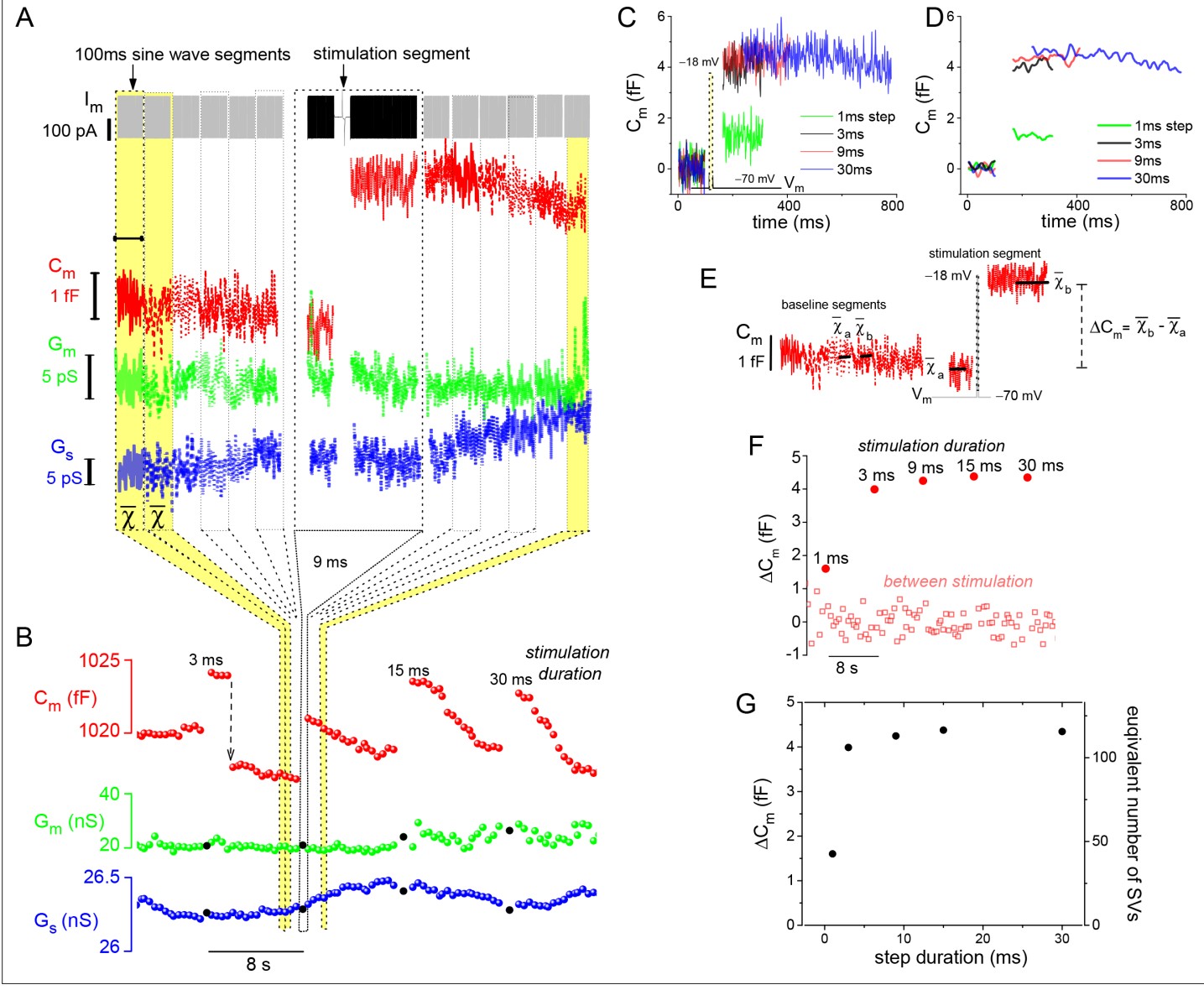

**Figure 2.** Resolution and analysis of evoked $\Delta C_m$. (**A**) The $I_m$ presented at the top of the figure is dominated by the sine-wave voltage protocol used to derive the lock-in amplifier outputs: $C_m$, $G_m$, and $G_s$. The protocol consisted of a series of 100 ms sine-wave segments presented in gray, which bracket the stimulation segment depicted in black (arrow points to the 9 ms depolarizing voltage step). (**B**) Responses to a series of step depolarizations for the indicated durations are presented. The approach used to bin the data in (**A**) and create the plot in (**B**) are illustrated schematically. The $G_m$ and $G_s$ data points that are filled in black mark the stimulation segments. The dashed, downward pointing arrow following the 3 ms stimulation highlights an unusually large endocytotic, downward step in $C_m$ (see text). (**C, D**) Plot the individual $C_m$ traces for each evoked response. Traces in (**D**) were low-pass filtered to a corner frequency ($f_c$) of 20 Hz, and in (**C**) a $f_c$ of 200 Hz was used. (**E**) Illustrates how $\Delta C_m$ was quantified over both baseline (between stimulation) and stimulation segments (see Materials and methods). (**F**) Chronological plot of $\Delta C_m$ during baseline and stimulation segments. (**G**) Summary plot with two y-axes: '$\Delta C_m$' and 'equivalent number of SVs,' versus step duration (x-axis). The conversion factor for calculating the number of SVs is described in Materials and methods. The $I_{Ca}$ traces measured from this cell are presented in **Figure 1C**. SV, synaptic vesicle.

In contrast to membrane endocytosis, the analysis of evoked membrane exocytosis was straightforward. An overlay of individual $C_m$ traces shows that the magnitude of the evoked $\Delta C_m$ responses were similar for step durations >1 ms (**Figure 2C and D**). To quantify stimulated and baseline changes in $C_m$, $\Delta C_m$ was estimated as indicated in **Figure 2E**, and then plotted in sequence over the course of the experiment (**Figure 2F**). Finally, the evoked $\Delta C_m$ was plotted per stimulation (pulse duration). In addition, the number of SVs corresponding to each $\Delta C_m$ was plotted on the opposite axis (**Figure 2G**),

which was calculated by dividing $\Delta C_m$ by the $C_m$ of a single SV: 37.6 aF (1 aF=$10^{-18}$ F; see Materials and methods for estimates of SV $C_m$); hence, a $\Delta C_m$ ~4 fF is equivalent to the fusion of 106 SVs.

## $Ca_v$ channels activate rapidly and exhibit moderate inactivation

The type of voltage-gated calcium channel that supports exocytosis from rod ribbons is the L-type, $Ca_v1.4$ channel. A feature that sets $Ca_v1.4$ channels apart from other L-type channels is that their expression is limited to PR terminals (for review, *McRory et al., 2004*; *Pangrsic et al., 2018*). By comparison, other L-type channels, such as $Ca_v1.2$ and 1.3, are expressed throughout the nervous

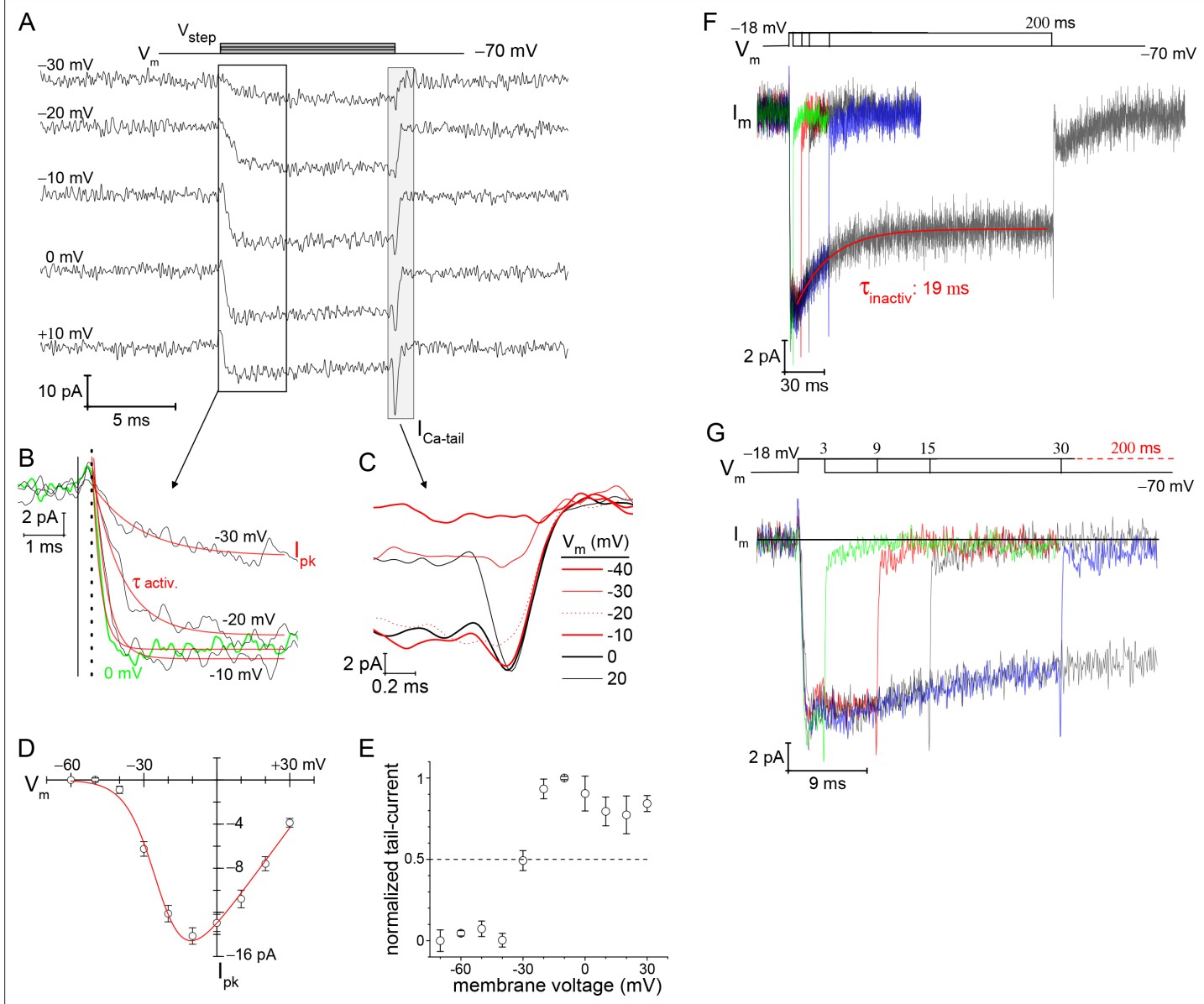

**Figure 3.** Voltage-dependence of $Ca^{2+}$-currents. (**A**) Presents a subset of individual $I_{Ca}$ traces taken from a rod filled with 10 mM EGTA. The 10 ms voltage steps were delivered in descending order from +30 to −80 mV at −10 mV increments, every 3 s. (**B**) Zoomed in view of $I_{Ca}$ activation. Start of the $V_{step}$ is indicated with the vertical solid line, and the onset of $I_{Ca}$ is indicated with a vertical dashed line. An exponential function was used to fit (red lines) each $I_{Ca}$ trace from which $\tau_{activation}$ and peak-$I_{Ca}$ were ascertained. (**C**) Overlay of tail currents from (**A**). (**D**) Averaged peak-$I_{Ca}$ versus $V_{step}$ fit with a modified Boltzmann I-V equation ($V_{1/2}$: −24.0±1.3, slope (dx): −6.2± 0.6 mV/e, $V_{rev}$: +44.6±2.6 mV, and $G_{max}$: 0.30±0.02 pA-mV$^{-1}$; 9 cells; see ***Supplementary files 2 and 3***). (**E**) Normalized $I_{Ca-tail}$ plotted over $V_{step}$ (4 cells). (**F, G**) Overlay of average membrane currents in response to voltage steps from −70 to −18 mV for the indicated durations (averages from 7 to 13 cells). The $I_{Ca}$ associated with the 200 ms step depolarizations (7 cells) were fit as a single exponential decay ($\tau_{inactivation}$). (**G**) Zoomed in view of the shorter duration voltage steps highlights the rapid return of $I_{Ca}$ to baseline (subsequent to the transient $I_{Ca-tail}$).

system, and additionally in cardiac, endocrine, and neuroendocrine cells, and such prevalence has led to significantly more insight into their biophysical properties (*Dolphin and Lee, 2020*). Therefore, some of the basic biophysical properties of mouse rod $Ca_v1.4$ channels, which have not been described to our knowledge, are reported next.

The voltage-dependence of $Ca^{2+}$-current activation was examined over a range of voltage steps when the rods were filled with 10 mM EGTA to minimize $Ca^{2+}$-activated $Cl^-$-currents (*Bader et al., 1982*). The depolarization stimulated $I_{Ca}$ showed a steep dependence on voltage (*Figure 3A*), and this was quantified by measuring peak-$I_{Ca}$ amplitude (*Figure 3B*) and calcium tail-current ($I_{Ca}$-tail) amplitude (*Figure 3C*). Specifically, a depolarizing $V_{step}$ to −40 mV produced a peak-$I_{Ca}$: −0.89±0.34 pA, and by −10 mV the maximal peak-$I_{Ca}$ was reached: −14.15±0.75 pA (9 cells; *Figure 3D*) (the liquid junction potential: $E_{lj}$=8.9 mV was not subtracted from $V_{step}$; see Materials and methods). Fitting the peak-$I_{Ca}$ versus $V_{step}$ curve with a Boltzmann equation from −60 to −10 mV (point of maximal $I_{Ca}$) gave a half-maximal peak-$I_{Ca}$ at a voltage of −28.7±0.4 mV (*Supplementary file 3*). Next, to better estimate the point when half of the available $Ca_v$ channels opened ($V_{1/2}$), a modified Boltzmann I-V equation was used, one that accounted for $V_{rev}$ and $G_{max}$ (see Materials and methods). This approach gave a $V_{1/2}$=−23.4±1.0 mV (fit presented in *Figure 3D*; see *Supplementary file 3* for additional statistics). Finally, $I_{Ca}$-tail amplitudes were measured to determine the fraction of channels that opened at each $V_{step}$, and this gave a half-maximal amplitude at approximately −30 mV (dashed line in *Figure 3E*). However, fitting the curve with a sigmoidal equation to estimate $V_{1/2}$ was not possible, because the $I_{Ca}$-tail amplitudes appeared to decrease at $V_{step}$ values positive to −10 mV (*Figure 3E*). This behavior may indicate a degree of $Ca_v$ channel inactivation within the 10 ms voltage step (see below). Therefore, the best estimate for half-maximal channel activation was derived from the modified Boltzmann I-V equation, and after subtracting $E_{lj}$, the $V_{1/2}$ is estimated to be ~ −33 mV.

Recordings from salamander rods have demonstrated that $I_{Ca}$ inactivation was absent in the presence of high concentrations (i.e., 10 mM) of intracellular EGTA, demonstrating an absence of voltage-dependent inactivation (VDI); however, lowering intracellular EGTA to ~0.1 mM gave rise to $Ca^{2+}$-dependent inactivation (CDI) (*Corey et al., 1984*; *Rabl and Thoreson, 2002*). To assess the situation in mouse rods, 200 ms steps were examined for signs of $I_{Ca}$ inactivation with 10 mM EGTA in the pipette. Steps from −70 to −18 mV, for varying durations, showed a decline in $I_{Ca}$ amplitude within the first 30 ms (*Figure 3F*). For instance, the 200 ms depolarizations had an initial peak-$I_{Ca}$=−13.2±0.8 pA, and ended with a mean current $I_{Ca}$=−9.2±1.2 pA (7 cells). This 31% decay in $I_{Ca}$ had a $\tau$=19.46±0.01 ms (7 cells; *Figure 3F*). The overlay of $I_{Ca}$ traces shows that the membrane current approached baseline within 1 ms after repolarization (*Figure 3G*), which is in accord with $Ca_v$ channel deactivation (*Figure 3C*), but not $Ca^{2+}$-activated tail-currents (see below). This result is distinct from previous studies on salamander rods that did not find evidence for VDI (*Bader et al., 1982*; *Corey et al., 1984*; *Rabl and Thoreson, 2002*). A notable difference in our study and that by *Corey et al., 1984* is that they replaced all intra- and extra-cellular monovalent cations with 100 mM $TEA^+$ to eliminate contributions from $K_v$ channels (*Beech and Barnes, 1989*). In contrast, we only used 20 and 35 mM $TEA^+$ inside and outside, respectively (see Materials and methods), which may not have been adequate to fully block $K_v$ channel currents. For these reasons, it is tentatively concluded that a fraction of the $Ca_v$ channels are sensitive to VDI.

## Expanding the $Ca^{2+}$-domain triggers $Ca^{2+}$-activated channels

It has been predicted that by changing intracellular EGTA from 10 to 0.5 mM, the $Ca^{2+}$-domain about the $Ca_v$ channels will expand in size from ~6 to ~ 210 nm, respectively, during depolarization (*Neher, 1986*). At ribbon synapses, high intracellular concentrations of EGTA (~10 mM) restrict the domain of evoked, free $Ca^{2+}$ to the base of the ribbon where the L-type $Ca_v$ channels are located (*Zenisek et al., 2003*; *Neef et al., 2018*; for review, *Moser et al., 2020*). A potential consequence of elevating intracellular $Ca^{2+}$ is that the $Ca^{2+}$-dependent $Cl^-$ currents ($I_{Cl(Ca)}$) will be activated (*Bader et al., 1982*). The TMEM16A/B channels are localized to mouse rod terminals (*Stöhr et al., 2009*; *Caputo et al., 2015*), and are thought to underlie $Ca^{2+}$-dependent $Cl^-$-currents in salamander PRs ($I_{Cl(Ca)}$) (*Mercer et al., 2011*). The above measurements made with 10 mM EGTA in the pipette did not exhibit a current resembling $I_{Cl(Ca)}$ (*Figure 3F*), which concurs with earlier studies on salamander and porcine rods that concluded $I_{Cl(Ca)}$ was blocked with high concentrations of EGTA (*Bader et al., 1982*; *Cia et al., 2005*). In contrast, when the intracellular concentration of EGTA was lowered to 0.5 mM, two new features

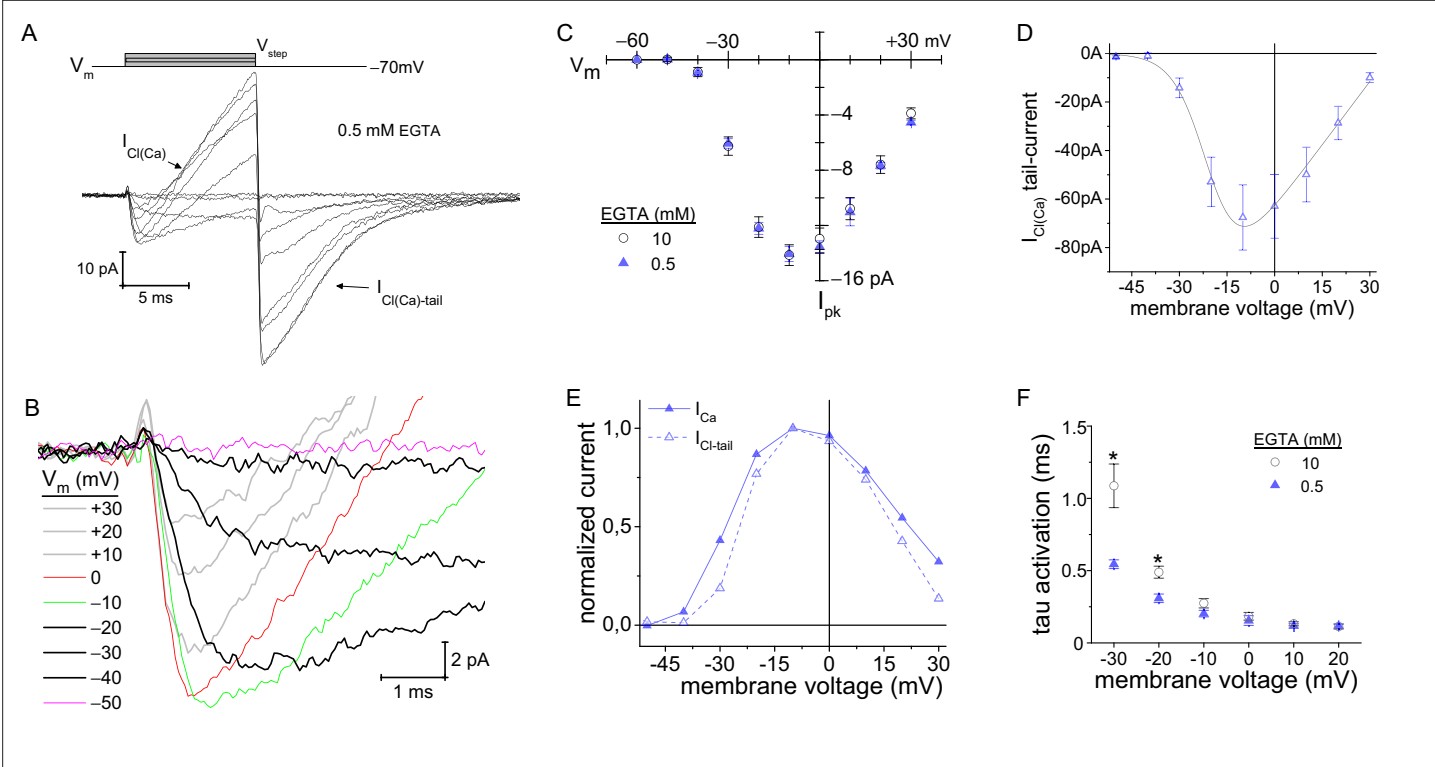

**Figure 4.** Lowering intracellular $Ca^{2+}$-buffering accelerated $Ca_v$ channel activation kinetics. (**A, B**) With 0.5 mM EGTA in the pipette, an outward current developed during the $V_{step}$ (designated as $I_{Cl(Ca)}$), which became an inward current upon repolarization ($I_{Cl(Ca)}$-tail). (**B**) Expanded view of the activation portion of the current traces from (**A**). Traces from 8 cells were averaged; depolarization protocol is described in *Figure 3A*. (**C**) Overlay of average peak-$I_{Ca}$ plotted over $V_{step}$ made from experiments with an intracellular concentration of either 0.5 (8 cells) or 10 mM EGTA . (**D**) Average $I_{Cl(Ca)}$-tails versus $V_{step}$ (6 cells). Modified Boltzmann I-V fit (blue trace): $V_{1/2}$: −20.4±0.5, slope (dx): −5.6±1 mV/e, $V_{rev}$: +35.8±3.8 mV and $G_{max}$: 1.41±0.44 pA-$mV^{-1}$ (6 cells; see *Supplementary file 4*). (**E**) Normalized average $I_{Cl(Ca)}$ and peak-$I_{Ca}$ from 0.5 EGTA data in (**C, D**) plotted over $V_{step}$ (error bars excluded). (**F**) Plot of $\tau_{activation}$ versus $V_{step}$ with an intracellular concentration of either 0.5 or 10 mM EGTA (*: p≤0.006; see *Supplementary file 2*).

appeared. First, an outward-current slowly developed during the 10 ms voltage step depolarizations (presumed $Cl^-$ influx). Second, at the end of the voltage step when the rod was repolarized to −70 mV, a inward tail-current appeared (presumed $Cl^-$ efflux), which slowly deactivated over the course of several milliseconds (*Figure 4A–B*). These features are indicative of a $Ca^{2+}$-dependent $Cl^-$ current (*Bader et al., 1982*; *Cia et al., 2005*).

If $I_{Cl(Ca)}$ is indeed $Ca^{2+}$-dependent, then it should follow the voltage-dependence of $I_{Ca}$. To evaluate this, first peak-$I_{Ca}$ had to be measured over the range of $V_{step}$ values, under the condition of low intracellular EGTA. The results show that peak-$I_{Ca}$ values were virtually identical for the two intracellular EGTA concentrations (*Figure 4C*; see *Supplementary file 2*); however, differences in activation kinetics were apparent (described below). Next, the peak-$I_{Cl(Ca)}$-tail currents were measured as a function of $V_{step}$. Since the $I_{Cl(Ca)}$-tail amplitudes were measured at −70 mV ($V_{rest}$), regardless of the preceding depolarizing $V_{step}$, the $I_{Cl(Ca)}$-tail amplitudes should be proportional to the number of TMEM16 channels opened; whereas, the amount of $Ca^{2+}$ entering depended on the $V_{step}$. The results show that peak-$I_{Cl(Ca)}$-tail and peak-$I_{Ca}$ currents followed a similar voltage-dependence (*Figure 4C and D*). This is best illustrated in the overlay of the normalized currents versus $V_{step}$, and here the $I_{Cl(Ca)}$-tail amplitudes scaled in proportion to $I_{Ca}$ over the entire range of $V_{step}$ values (*Figure 4E*; see *Supplementary file 4* for Boltzmann fits). This demonstrates that the $Ca^{2+}$-dependent current activated in proportion to the amplitude of peak-$I_{Ca}$. In summary, 10 mM EGTA blocked activation of the $Cl^-$-channels in mouse rods, as has been described previously in studies on salamander and porcine rods (*Bader et al., 1982*; *Cia et al., 2005*; *Mercer et al., 2011*), which suggests the $Ca^{2+}$-activated $Cl^-$-channels are not localized within nanometers of the $Ca_v$ channels.

## Reducing intracellular EGTA accelerates $Ca_v$ channel activation kinetics

It is known that cytoplasmic $Ca^{2+}$ can facilitate the opening of $Ca_v$ channels (*Lee et al., 2000*; *Nanou and Catterall, 2018*). This behavior has been documented at rat inner hair cell (IHC) ribbon synapses (*Grant and Fuchs, 2008*; *Goutman and Glowatzki, 2011*) and at central synapses (*Borst and Sakmann, 1998*), but it is not known if mammalian PRs support similar channel dynamics. Interestingly, when comparing results from experiments with 0.5 and 10 mM EGTA in the pipette, the activation kinetics were slowed to a greater extent at negative voltages when 10 mM EGTA was used. Specifically, with 10 mM EGTA, the time constant for $I_{Ca}$ activation ($\tau_{activ}$) was lengthened sixfold when changing the $V_{step}$ from 0 to −30 mV (0.18±0.03 to 1.09±0.15 ms, respectively; 9 cells; *Figure 4F*), compared to the 3.4-fold lengthening of $\tau_{activ}$ when 0.5 mM EGTA (0.16±0.03 to 0.55± 0.03 ms, respectively; 8 cells; *Figure 4F*). Furthermore, in the physiological range for $Ca_v$ channel activation, $\tau_{activ}$ was twofold faster at −30 mV (−40 mV after correction for $E_{lj}$) when 0.5 mM EGTA was used (p: 0.003; *Figure 4F*). Comparison of activation kinetics for low (0.5 mM) and high (10 mM) EGTA shows that they converged at more depolarized voltages (*Figure 4F*; see *Supplementary file 2*). Assuming basal intracellular $Ca^{2+}$ was higher when less EGTA was used, then the results can be interpreted as $Ca^{2+}$-dependent facilitation of $Ca_v$ channel activation kinetics (*Borst and Sakmann, 1998*); but not peak-$I_{Ca}$ amplitude (*Figure 4C*). The results bare some resemblance to the behavior of $Ca_v1.3$ channels found in rat IHCs (*Goutman and Glowatzki, 2011*), in that the time to peak-$Ca^{2+}$ current is shortened when residual intracellular $Ca^{2+}$ is elevated; however, current onset delay was shortened in the case of IHCs.

## The readily releasable pool of SVs is primed for ultrafast release

As outlined earlier in *Figure 1*, evoked $\Delta C_m$ can be related to the number of SVs that fused with the plasma membrane. We first used this approach to determine how many SVs were near $Ca_v$ channels and ready for release (*Mennerick and Matthews, 1996*; *Moser and Beutner, 2000*; *Singer and Diamond, 2003*; *Thoreson et al., 2004*; *Graydon et al., 2011*). This was achieved by using 10 mM EGTA in the pipette to restrict the evoked $Ca^{2+}$-domain to within nanometers of the $Ca_v$ channels, and a series of brief depolarizing voltage steps were given to map out the initial phase of release ($V_{rest}$=−70 mV, and $V_{step}$=−18 mV; ordering of steps: 0.5, 1, 3, 9, 15, and 30 ms, with 8 s of rest between stimulations). Inspection of the plot of $\Delta C_m$ versus step duration shows that the greatest change occurred within the first 3 ms of depolarization (*Figure 5A*). This point is documented in a couple of ways. First, the $\Delta C_m$ generated with 3 ms steps reached 86% the amplitude of responses evoked with a 30 ms step (2.99±0.48 vs. 3.54±0.50 fF; p: 0.033, paired sample t-test; 7 cells; *Figure 5A*). The second approach involved making a comparison of exponential fits ($\tau_{depletion}$) to the average $\Delta C_m$ versus step duration over different ranges of stimulations. Specifically, a fit from 0.5 to 9 ms gave a $\tau_{depletion}$=348 µs and an amplitude=3.27 fF (adjusted R-square=0.999), which was similar to the result attained when the fit was extended out to 30 ms: $\tau_{depletion}$=383 µs and amplitude=3.43 fF (adjusted R-square=0.992; 7–15 cells per step duration; *Figure 5A*). This suggests the initial kinetic release phase, referred to here as the fusion of the RRP of SVs, expired with a time constant <0.4 ms and amounted to 3.27 fF (~87 SVs). It is worth noting that only slightly more time was needed to empty the RRP than indicated by $\tau_{depletion}$. The 0.5 ms steps produced negligible changes in $\Delta C_m$ (0.14±0.19 fF ~4 SVs; 8 cells); whereas, the 1 ms steps released ~ 70% of the RRP ($\Delta C_m$: 2.11±0.74 fF ~56 SVs; 14 cells). This indicates a fusion delay of ~0.5 ms, involving $Ca_v$ channel activation kinetics and $Ca^{2+}$-dependent activation of SV fusion, which was followed by ultrafast depletion of the RRP in under a millisecond. Similarly rapid depletion of the RRP of SV formed at goldfish Mb1 bipolar terminals (*Mennerick and Matthews, 1996*; *Palmer et al., 2003*) and rodent calyxes of Held (*Schneggenburger et al., 2002*) have been reported when using strong step depolarizations.

Before advancing to the next set of results, a description of the experimental design is given. The first expectation was that the RRP of SVs refilled in <8 s, which was the inter-stimulus interval used in the short step duration protocols. The second assumption was that evoked responses adapted over the cumulative time course of the experiment. To the first point, AMPA receptor-mediated EPSCs recorded from ground squirrel cb2 bipolars, paired to either a presynaptic rod or green-cone, recovered from paired-pulse depression within 0.3 s (*Li et al., 2010*). However, longer recovery times of ~1 s have been reported for green-cones paired to cone bipolars that express a majority of kainate receptors (*DeVries, 2000*). Finally, presynaptic release ($\Delta C_m$) measured from green-cones recovered

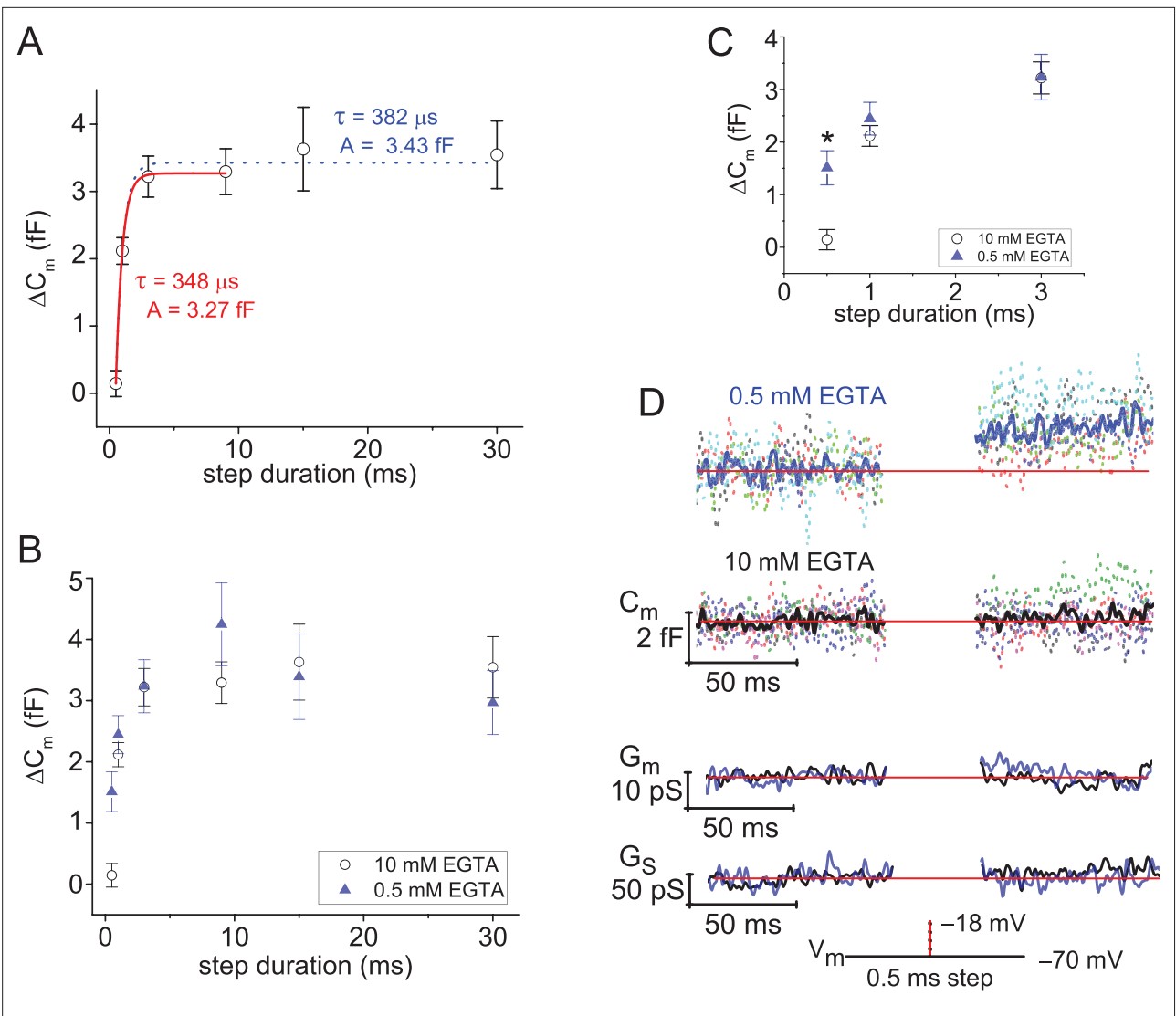

**Figure 5.** Ultrafast depletion of the RRP of SVs. (**A**) Average $\Delta C_m$ measured from rods filled with 10 mM EGTA and stimulated with a $V_{step}$ to −18 mV for durations from 0.5 to 30 ms. Single exponential fits to points from 0.5 to 9 ms (red curve), and from 0.5 to 30 ms (dotted curve). Stimulations were delivered in ascending order, and only a single cycle per cell. (**B**) Comparison of $\Delta C_m$ derived from experiments with 0.5 or 10 mM EGTA in the pipette; voltage step durations: 0.5–30 ms. (**C**) Highlights the more rapid $\Delta C_m$ at the singular time point of 0.5 ms when rods were loaded with 0.5 mM EGTA (*, p: 0.0016; 6 and 8 cells for 0.5 and 10 mM EGTA, respectively). (**D**) Summary of lock-in amplifier traces recorded during 0.5 ms step depolarizations with either 0.5 or 10 mM EGTA in the pipette (6 and 8 cells, respectively). Dashed $C_m$ traces represent an overlay of individual recordings (cells; each a different color), and the $C_m$ trace presented in bold font represents the average response. Only an overlay of the average responses in low and high EGTA are presented for $G_m$ and $G_s$ with the blue traces corresponding to 0.5 mM EGTA and black traces corresponding to 10 mM EGTA. RRP, readily releasable pool; SV, synaptic vesicle.

from paired-pulse depression in 0.7 s (*Grabner et al., 2016*). Without additional studies on mammalian PRs to point to, we note that salamander rods and cones recovered from paired-pulse depression (assayed as EPSCs or $\Delta C_m$) with a $\tau_{recovery}$ ~1 s or less (*Rabl et al., 2006*; *Innocenti and Heidelberger, 2008*). On this backdrop, one can assume that 8 s was enough time for mouse rods to recover their RRP of SVs. Furthermore, the data plotted in *Figure 5A*: $\Delta C_m$ versus step duration, show a drastic difference between the first two stimulations given at 0.5–1 ms, but subsequent, longer duration stimulations (3–30 ms) showed only a modest incremental increase in $\Delta C_m$. This is interpreted as depletion of a finite pool of primed SVs (*Mennerick and Matthews, 1996*). Based on the literature and data, a stimulation interval <8 s would be justified; however, we also had to accommodate experiments on *Ribeye*-ko mice, described below, that may need more time to recover from stimulation. In addition,

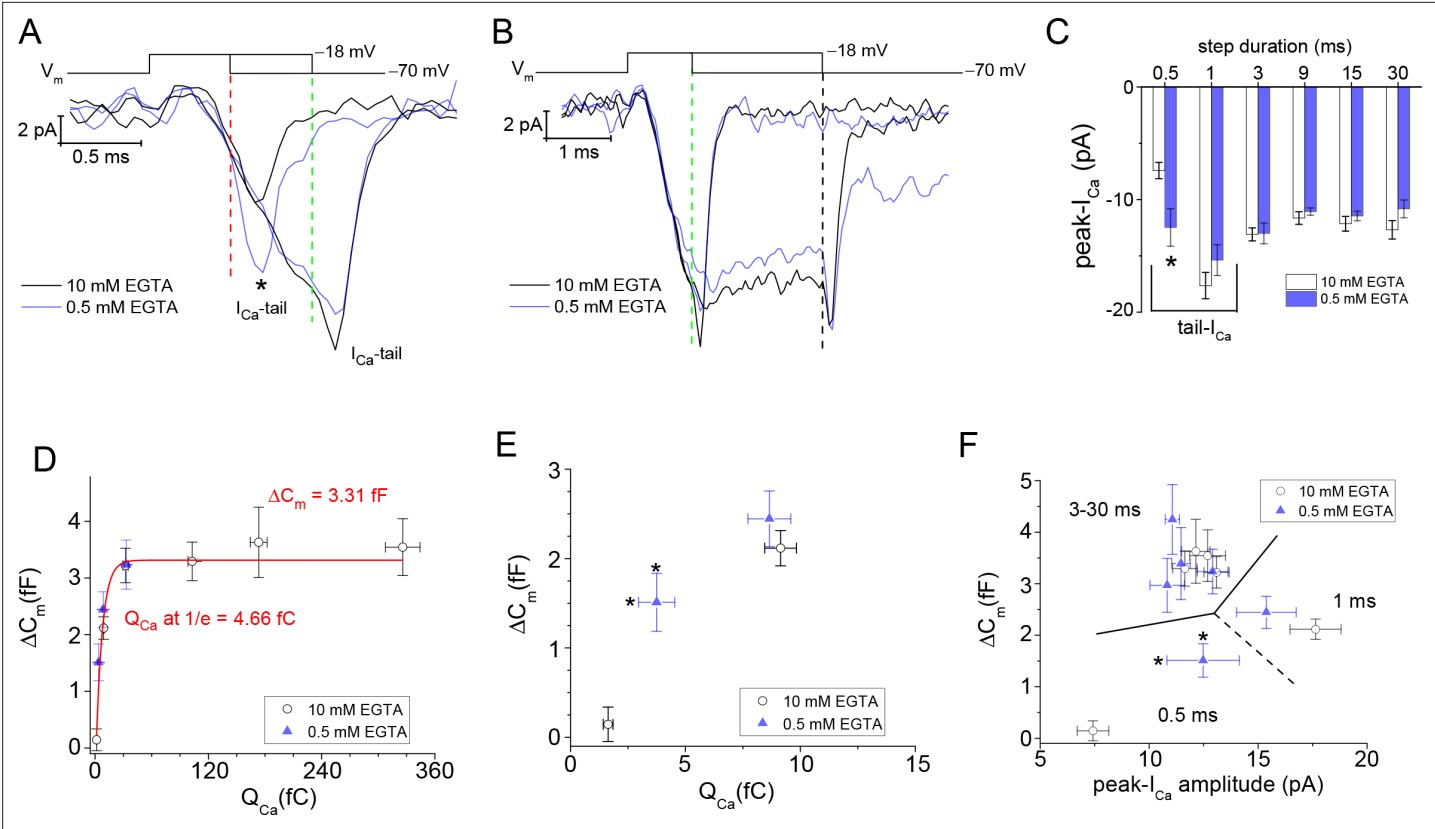

**Figure 6.** $I_{Ca}$ facilitation expedites depletion of the RRP. (**A, B**) Overlay of average $I_{Ca}$ traces in response to step durations of 0.5, 1, and 3 ms (*; sign. diff., p: 0.008). Vertical dashed lines indicate the moment of repolarization. (**C**) Comparison of average peak-$I_{Ca}$ derived from high and low intracellular EGTA. Only voltage steps for 0.5 ms were statistically different when comparing different EGTA levels (p: 0.008; see text). The 0.5 and 1 ms steps were dominated by their tail-currents, which is why their amplitudes varied from step durations ≥3 ms. (**D**) Plot of $\Delta C_m$ over $Q_{Ca}$. All data points from experiments with 0.5 and 10 mM EGTA were treated as one group and fit with a single exponential equation (red curve) to estimate the size of the RRP ($\Delta C_m$ amplitude) and the amount of $Q_{Ca}$ needed to release 63% (~1/e) of the RRP. (**E**) Plot of $\Delta C_m$ and $Q_{Ca}$ values produced with 0.5 and 1 ms steps. * indicates a significant difference for $Q_{Ca}$ between 0.5 mM versus 10 mM EGTA; p: 0.04, and 6 and 9 cells, respectively. (**F**) Plot of $\Delta C_m$ over peak-$I_{Ca}$ for step durations: 0.5–30 ms (dashed lines partition data points by step duration). * indicates a significant difference in $I_{Ca}$ that is described in (**C**), and differences in $\Delta C_m$ are described in *Figure 5C*. RRP, readily releasable pool.

experiments with an intracellular concentration of 0.5 mM EGTA may need more time to clear intracellular $Ca^{2+}$ between stimulations (*Van Hook and Thoreson, 2015*).

To address the second concern: time-dependent adaptation after whole-cell entry, the overall experimental time was minimized. This effort entailed starting stimulations ~ 35 s after gaining whole-cell access, and the protocol was circumscribed to mapping out the RRP of SVs (the stimulation protocol is described above). This protocol amounted to less than 2 min from the time of entry. Furthermore, an infusion time of 35 s was sufficient to affect $Ca^{2+}$-buffering, because $Ca^{2+}$-activated currents only appeared when 0.5 mM EGTA was in the pipette (*Figure 4A* versus *Figure 3F*). This conclusion was reinforced in the experiments presented in *Figure 6A and B*, and described below. These features suggest that rod terminals were adequately infused. For reference, a comparison to experiments performed on mouse rbcs is made. The rbc forms a single axon that is ~60 μm long, and it branches into multiple terminal synapses. *Singer and Diamond, 2003* showed that after gaining whole-cell access via the rbc soma, synaptic terminals were filled with exogenous $Ca^{2+}$-buffers within 2 min (*Singer and Diamond, 2003*).

## Lowering the intracellular $Ca^{2+}$-buffer expedites release onset

To assess whether primed SVs resided outside the spatially constrained $Ca^{2+}$-domain formed with 10 mM EGTA, as has been proposed to occur at goldfish bipolar AZs (*Burrone et al., 2002*), rods were filled with 0.5 mM EGTA and given a series of short depolarizing steps (as described above).

*Figure 5B* plots $\Delta C_m$ versus step duration for experiments carried out with either 0.5 or 10 mM EGTA in the pipette, and the results show a high degree of overlap within the range from 1 to 30 ms steps. In contrast, $\Delta C_m$ in response to 0.5 ms step depolarizations were more than tenfold larger when rods were filled with 0.5 mM versus 10 mM EGTA (1.51±0.36 vs. 0.14±0.19 fF; p: 0.006; n: 6 and 8 cells; *Figure 5C*). This is also illustrated in the plots of individual $C_m$ traces measured in response to 0.5 ms steps (*Figure 5D*). The results show that the size of the RRP of SVs was comparable under high and low $Ca^{2+}$-buffering, but the RRP emptied faster with less intracellular EGTA. The fixed size of the RRP with 0.5 or 10 mM EGTA in the pipette is compatible with studies on mouse rod bipolars *Singer and Diamond, 2003*; whereas, the accelerated rate of release was highly reminiscent of goldfish bipolars (*Burrone et al., 2002*). To better understand this, we next considered whether the $Ca^{2+}$-currents were differentially impacted by the two $Ca^{2+}$-buffering conditions.

## Facilitation of Ca$_v$ channel activation kinetics expedites release onset

From the $I_{Ca}$ traces used to profile the voltage-dependence of Ca$_v$ channel activation kinetics (*Figures 3B and 4B*), and those associated with evoked release experiments (*Figure 6A and B*), a delay of ~300 µs existed prior to the onset of the $I_{Ca}$. Hence, in instances where 0.5 ms voltage steps were delivered to evoke release, only ~200 µs were available for the $I_{Ca}$ to develop before the voltage step ended. Given that $\tau_{activ}$ at −20 mV for low and high EGTA were 310 µs versus 489 µs, respectively (*Figure 4F*, and *Supplementary file 2*), it is expected that more channels opened with less $Ca^{2+}$-buffering during the 0.5 ms steps. The results from evoked release show that the average $I_{Ca}$-tail resulting from 0.5 ms steps to −18 mV were approximately twofold larger when less EGTA was used (0.5 vs. 10 mM EGTA: −12.5±1.7 vs. −7.4±0.7 pA; p: 0.008 from 6 and 9 cells; *Figure 6A and C*). In contrast, the 1 ms steps produced comparable $I_{Ca}$-tail amplitudes under the two intracellular EGTA conditions (*Figure 6A–C*), because the step duration was > $\tau_{activ}$. Likewise, step depolarizations for 3 ms and longer gave similar peak-$I_{Ca}$ amplitudes that ranged from −11 to −13 pA in both low and high $Ca^{2+}$-buffering (*Figure 6C*). Since the $Ca^{2+}$-buffering conditions used to evoke exocytosis differentially facilitated $Ca^{2+}$-entry within the timeframe of Ca$_v$ channel activation, the next analysis evaluated $\Delta C_m$ as a function of the amount of $Ca^{2+}$ that entered.

First, a plot of $\Delta C_m$ versus the integral of $I_{Ca}$ ($Q_{Ca}$) was made from a combination of experiments with low and high intracellular EGTA (11 and 15 cells, respectively). The outcome was a continual, single exponential process that reached an amplitude of 3.31 fF (RRP ~88 SVs; *Figure 6D and E*), and 63% of the RRP was depleted (1/e) when $Q_{Ca}$ reached 4.66 fC=12,263 $Ca^{2+}$ ions (1 $Ca^{2+}$ ion=3.8 × 10$^{-19}$C) (*Figure 6D*). A limitation here was that $Q_{Ca}$ was not calculated for steps >3 ms when 0.5 mM EGTA was used, because $I_{Cl(Ca)}$ interfered with determination of $Q_{Ca}$. The second comparison made here was $\Delta C_m$ versus $I_{Ca}$ amplitude. This is an alternative to $Q_{Ca}$ for indexing $Ca^{2+}$ entry, and since peak-$I_{Ca}$ does not appear to be impacted by $I_{Cl(Ca)}$ (*Figure 4C*), all of the peak-$I_{Ca}$ amplitude data points from experiments with low EGTA were included in *Figure 6F*; and a combination of $I_{Ca}$-tail (from 0.5 and 1 ms steps) and peak-$I_{Ca}$ (steps≥3 ms) were plotted against $\Delta C_m$. A simple linear or exponential process was not realized; but instead, the data points were scattered into groups. The steps≥3 ms were grouped within a range of $\Delta C_m$ values between 3 and 4 fF (equivalent to the RRP), and with a peak-$I_{Ca}$ of approximately −12 pA (*Figure 6F*). The remaining data points were generated with 0.5 and 1 ms steps, and they produced a broad range of $\Delta C_m$ values (0.2–2 fF) and $I_{Ca}$-tail amplitudes (−7.5 to −18 pA) that suggest $\Delta C_m$ scaled with $I_{Ca}$-tail amplitude (*Figure 6F*).

In summary, plotting $\Delta C_m$ against step duration demonstrates ultrafast depletion of an RRP of ~90 SVs in response to a moderate amount of $Ca^{2+}$ entry (~10,000 ions). These features point to a critical role for Ca$_v$ channel activation kinetics, and can be interpreted as a primed SV fusing once its neighboring channel(s) opens (*Jarsky et al., 2010*). To this point, our experiments with low intracellular EGTA led to the majority of Ca$_v$ channels activating in ~0.6 ms ($I_{Ca}$ onset delay: 300 µs plus the $\tau_{activ}$: 310 µs; *Figures 4C, 6A and B*; *Supplementary file 2*). This length of time was all that was needed to fuse the majority of the RRP of SVs (*Figure 5B and C*). Since 10 mM EGTA slowed Ca$_v$ channel activation kinetics, we propose that the small time delay in $Ca^{2+}$ entry is what produced a delay in exocytosis. For comparison, changing intracellular EGTA levels has been shown to accelerate the onset of $Ca^{2+}$-entry and in turn accelerated release kinetics (*Borst and Sakmann, 1998*; *Goutman and Glowatzki, 2011*). However, it is also expected that the different EGTA levels will significantly alter the size of the $Ca^{2+}$-domain formed about Ca$_v$ channels. This raises the possibility that a larger

$Ca^{2+}$-domain might access additional releasable SVs, which release at different rates (*Neher and Brose, 2018*); however, the size of the RRP was not influenced by the two intracellular EGTA concentrations used in our study. For these reasons, we conclude that rods use a $Ca^{2+}$-nanodomain that tightly couples $Ca_v$ channels and SVs to control their release (reviewed here: *Moser et al., 2020*).

Since high-resolution immuno-fluorescence imaging showed that $Ca_v$ channels were concentrated at the base of the rod ribbon (*Dembla et al., 2020*), we assume that this is where the RRP of SVs are docked. From electron microscopy (EM) studies on mouse rod ribbons, the number of SVs estimated to be docked at the base of the ribbon range from 60 to 86 (*Zampighi et al., 2011*; *Cooper et al., 2012*; *Grabner et al., 2015*); thus, approximately the size of the RRP of 87 SVs we have measured in this study. An additional 300 SVs reside near the plasma membrane along the synaptic ridges (*Zampighi et al., 2011*; for review, *Moser et al., 2020*), which significantly exceeds the size of the RRP SVs.

## Ribbonless rods support only marginal evoked exocytosis

An earlier EM study, which introduced the *Ribeye*-ko mice, showed that the ribbonless rod AZs maintained 60% fewer SVs than wild type rod ribbon AZs (*Maxeiner et al., 2016*). This former study did not measure release from rods; therefore, we tested whether the loss of SVs from the AZ would affect the size of the RRP of SVs. The measurements were first performed with 10 mM EGTA in the pipette. The evoked responses recorded from ribbonless rods given 1 and 3 ms $V_{step}$ to −18 mV amounted to 0.21±0.20 and 0.68±0.34 fF, respectively (*Figure 7A*; 7 and 8 cells). These responses were ten- and fivefold smaller than wt $\Delta C_m$ evoked with 1 and 3 ms steps: 2.12±0.20 and 3.22±0.31 fF (*Figure 7B*), respectively (p-values<0.0001 for wt vs. ko; 14 and 15 cells). Responses from ribbonless rods elicited with longer step durations (out to 30 ms) did not exceed the $\Delta C_m$ elicited by 3 ms steps (*Figure 7A*). Fitting $\Delta C_m$ versus step duration with a single exponential equation gave a $\tau_{depletion}$=560 µs and a $\Delta C_m$ amplitude=0.84 fF (RRP ~22 SVs; 8 cells), which is approximately 24% the size of that measured in wt rods ($\Delta C_m$: 3.43 fF; *Figure 7A*).

To further investigate if the reduction in exocytosis measured from ribbonless rods reflected a reduction in the number of SVs available for release or an impairment in the coupling of $Ca^{2+}$ influx to SVs, as concluded to occur at ribbonless retinal bipolar cells (*Maxeiner et al., 2016*), the concentration of intracellular EGTA was lowered. Ribbonless rods filled with 0.5 mM EGTA and given 1 and 3 ms stimulations produced $\Delta C_m$ values that averaged 0.70±0.16 and 0.86±0.67 fF, respectively (*Figure 7B*; 12 cells for each group). In contrast, when wt rods filled with 0.5 mM EGTA were given 1 and 3 ms stimulations, $\Delta C_m$ responses average 2.44±0.31 and 3.24±0.43 fF, respectively (*Figure 7B*; 11 cells in each group); thus, ribbonless rods generated evoked $\Delta C_m$ that were < 30% the size of wt responses (p-values for wt vs. ko, for 1 and 3 ms: 0.00016 and 0.00023). Overall, the results from the different $Ca^{2+}$-buffering conditions indicate that the RRP formed by ribbonless rods was only ~22 SVs. Interestingly, the different EGTA concentrations did not significantly influence RRP size within either genotype, nor were the kinetics of depletion altered. Thus, these results suggest that the coupling of $Ca^{2+}$ influx to exocytosis was not noticeably different between genotypes, but rather the number of release ready SVs distinguished wt from ribbonless rods.

## Deleting the ribbon alters $Ca_v$ channel properties

Previous work had indicated the density of $Ca_v$1.4 (α1F subunit) staining was altered in ribbonless rods (*Maxeiner et al., 2016*; *Dembla et al., 2020*). More to the point, the ribbon-shaped profile that $Ca_v$1.4 channels adhere to in wt rods was reduced in length by 50% in ribbonless rod terminals; however, protein levels of the $Ca_v$1.4 α1F subunit examined with Western blots were found to be similar in wt and ko retina (*Maxeiner et al., 2016*). To test if these changes affected the behavior of $Ca_v$1.4 channels, a comparison of $Ca^{2+}$ currents from wt and ribbonless rods was made. Results derived from 10 ms voltage steps showed a significant overall reduction in the ribbonless rods peak-$I_{Ca}$ over the range of $V_{step}$'s from −30 to +30 mV (*Figure 7C*, and see *Supplementary files 2 and 3*). Specifically, with 10 mM EGTA in the pipette, the peak-$I_{Ca}$ amplitude was approximately 35% smaller in ko rods than in wt rods (p<0.001 or smaller, depending on $V_{step}$; 9 wt and 5 ko cells), and when 0.5 mM EGTA was used the amplitudes were 20% smaller than in wt controls (p<0.05 or less; 8 wt and 7 ko cells; *Figure 7C*, and *Supplementary file 3*). Additional biophysical values were derived from Boltzmann fits to the I-V curves. This analysis shows that ko rods had a maximal conductance ~ 34% smaller

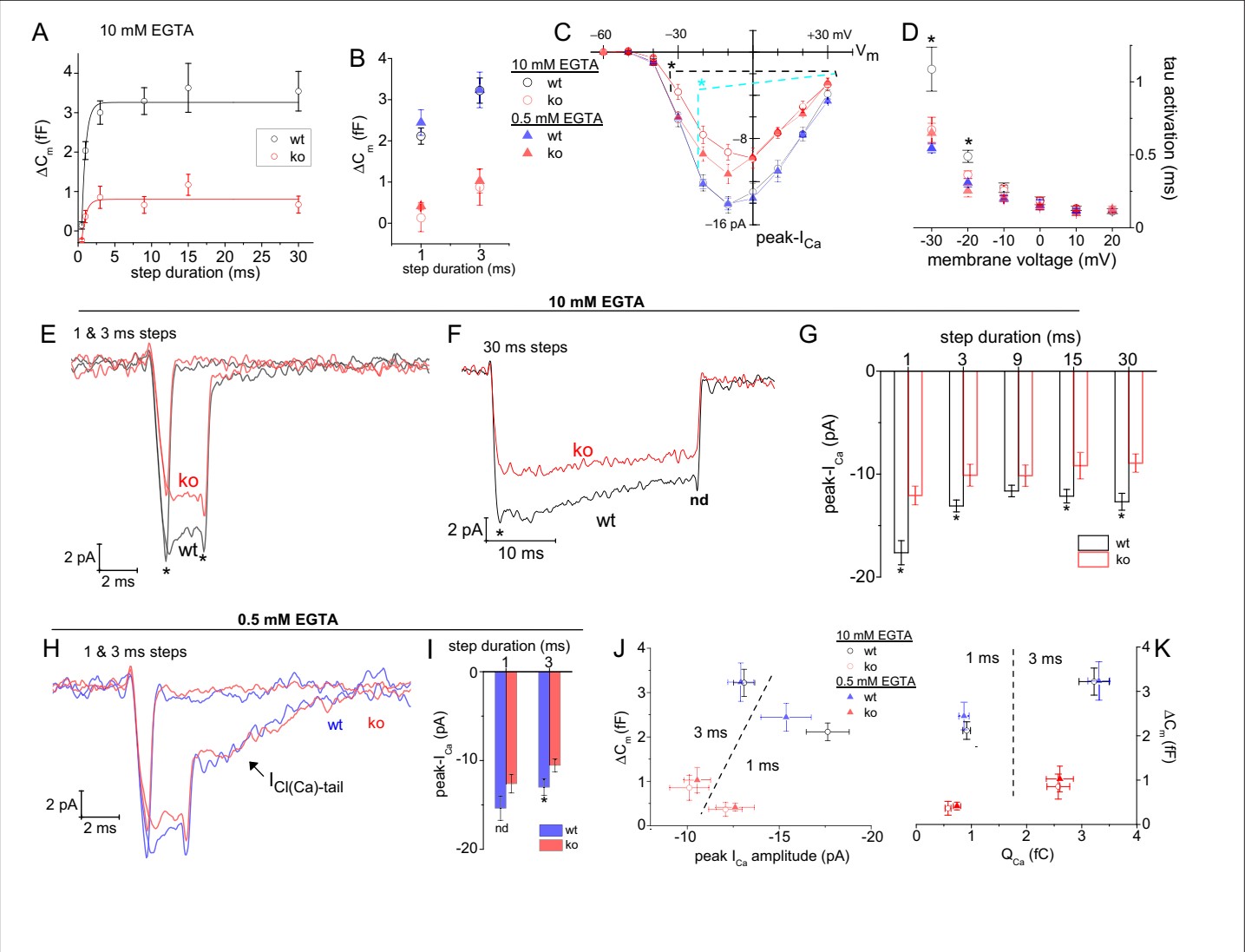

**Figure 7.** Ribbonless rods lack Ca$_v$ channel facilitation and form a small RRP of SVs. (**A**) Average $\Delta C_m$ measured from ribbonless rods filled with 10 mM EGTA and stimulated with a V$_{step}$ to −18 mV for durations from 0.5 to 30 ms. Single exponential fits to points from 0.5 to 30 ms. Wild-type results are presented for comparison (see *Figure 5A*). (**B**) Comparison of all $\Delta C_m$ responses that were evoked with 1 and 3 ms steps, with either 0.5 or 10 mM EGTA in the pipette, and for wt and ko rods. All ko responses were significantly smaller than wt values, see text. (**C**) Plot of peak-I$_{Ca}$ over V$_{step}$. Ribbonless rods had significantly smaller peak-I$_{Ca}$ at the indicated V$_{step}$ values (*, p-values<0.05), with comparisons made between wt vs. ko rods for 0.5 mM EGTA (blue dashed line) and 10 mM EGTA (black dashed line). See *Figure 3A* for a description of voltage step protocols and analysis. (**D**) I$_{Ca}$ activation kinetics were significantly slower for wt rods filled with 10 mM EGTA (* indicates sig. diff.; see *Supplementary file 2*). (**E, F**) Average I$_{Ca}$ traces measured from rods filled with 10 mM EGTA illustrate the significant differences in peak-I$_{Ca}$ at the onset of depolarization (*; p<0.004 and 0.03 for 1 and 3 ms steps; n: 13 wt and 7 ko cells). See *Supplementary file 3* for related results with 10 ms steps. (**F**) wt rods showed a faster rate of I$_{Ca}$ inactivation than ko rods. At the end of the 30 ms steps, I$_{Ca}$ amplitudes were no longer significantly different (nd; p>0.3); whereas, at the onset they were different (*; p<0.008; n: 8 wt and 7 ko). *Supplementary file 3* (**G**) Summary of average peak-I$_{Ca}$ measured from wt and ko rods filled with 10 mM EGTA. The 1 ms steps were essentially tail-currents, and therefore larger in amplitude than peak-I$_{Ca}$ measured from step durations ≥3 ms. Only the 9 ms steps did not show a significant difference between wt and ko. (**H, I**) Comparison of I$_{Ca}$ recorded with 0.5 mM EGTA in the pipette. Averaged traces presented in (**H**). In (**I**), statistical comparisons show no difference in I$_{Ca}$ amplitudes resulting from 1 ms steps (nd; p: 0.13; 9, and 11 cells each), while the currents generated with 3 ms steps were significantly larger for wt rods (*, p: 0.05; 9 and 12 cells each). (**J, K**) $\Delta C_m$ in response to 1 and 3 ms steps plotted over peak-I$_{Ca}$ in (**J**), and over Q$_{Ca}$ in (**K**). Dashed lines partition data points by step duration. See text for discussion. RRP, readily releasable pool; SV, synaptic vesicle.

The online version of this article includes the following figure supplement(s) for figure 7:

**Figure supplement 1.** Average I$_{Cl(Ca)}$-tails versus V$_{step}$.

than wt values, and this was true for experiments performed with high and low intracellular EGTA concentrations (p: 0.024 and 0.029, respectively; *Supplementary file 3*); however, $V_{1/2}$ for peak-$I_{Ca}$ amplitude was not significantly changed with high or low EGTA (p ~ 0.17; 8 and 7 cells; *Figure 7C* and see *Supplementary file 3*). The final comparison made from the voltage steps was $I_{Ca}$ activation kinetics. Currents activated at similar rates, except for wt rods filled with 10 mM EGTA, which exhibited much slower activation kinetics (*Figure 7D* and *Supplementary file 3*).

The reduction in peak-$I_{Ca}$ was also observed in recordings that depolarized rods to −18 mV for different durations with 10 mM EGTA in the intracellular solution (*Figure 7E–G*); however, over time the difference faded as $Ca_v$ channels in wt rods inactivated more rapidly. Specifically, 30 ms steps had an initial peak-$I_{Ca}$ that was  35% larger in wt rods (−13.3±0.43 pA vs. ko: −8.7±0.8 pA; p: 0.0002; 7 per genotype; *Figure 7D*), but by the end of the 30 ms step, the wt and ribbonless $I_{Ca}$ were no longer statistically different (−8.7±0.8 pA vs. −7.4±1.1 pA, p: 0.35; 7 cells each; *Figure 7F*). In the case of recordings with 0.5 mM EGTA in the pipette, the 3 ms depolarizations also produced slightly smaller $I_{Ca}$ amplitudes in the ribbonless rods (*Figure 7H*, and see *Figure 7I* for statistical comparisons). In total, $Ca_v$ channel behavior in ribbonless rods were distinguished from their wt counter parts in one or more of the following ways: they had a lower $I_{Ca}$ density at the onset of depolarization, and they exhibited less current inactivation (or less facilitation at onset).

The influence of peak-$I_{Ca}$ and $Q_{Ca}$ on evoked $\Delta C_m$ were also summarized. The responses to 1 and 3 ms steps, with low or high intracellular EGTA are plotted for both genotypes in *Figure 7J and K*. These plots show that the ribbon greatly influenced exocytosis.

## Ca²⁺-activated currents are reduced in ribbonless rods

Ribbonless rods $I_{Cl(Ca)}$-tail amplitudes were  60% smaller than wt rods (*Figure 7—figure supplement 1*; *Supplementary file 4*). This parallels the observation that ribbonless rods peak-$I_{Ca}$ amplitudes were significantly smaller than wt rods (*Supplementary file 3*). Next, the voltage for half-maximal $I_{Cl(Ca)}$ was reached at more negative membrane voltages than wt rods ($V_{1/2}$ for $I_{Cl(Ca)}$), wt vs. ko: −20.4±0.5 vs. −22.8±0.6 mV, respectively; p: 0.016 (*Supplementary file 4*). In spite of these differences, the overall Ca²⁺-dependence of $I_{Cl(Ca)}$ appeared similar for wt and ribbonless rods. This is illustrated in the plot of normalized $I_{Cl(Ca)}$ and normalized peak-$I_{Ca}$ over $V_{step}$, which shows the $I_{Cl(Ca)}$–$V_{step}$ curves are bounded by the peak-$I_{Ca}$–$V_{step}$ curves (*Figure 7—figure supplement 1*).

## Light responses in ribbonless mice are greatly reduced

The results so far show a stark reduction in the RRP of SVs. In the intact animal, this deficit is expected to preferentially impact rod signaling in the dark, a period when the rate of glutamate release is the highest. Normally, presenting a dim light flash to a dark-adapted retina will cause a momentary pause in glutamate release from rods, which in turn causes a depolarization (dis-inhibition) of postsynaptic rbcs. The magnitude of the light response, as assessed with electroretinogram (erg) recordings, will reflect the extent of rbc depolarization. If *Ribeye*-ko rods are unable to keep synaptic glutamate levels high enough to inhibit rbcs in the dark, then their scotopic-erg light responses should reflect this. To test this hypothesis, ergs were performed on dark-adapted mice, under scotopic test conditions (*Figure 8A–B*). The amplitude of the erg a-wave reflects phototransduction in the outer segments. Wild-type and *Ribeye*-ko mice had similar a-wave amplitudes across the range of flash intensities (*Figure 8C*). In contrast, the *Ribeye*-ko mice generated erg b-wave amplitudes that were only 52– 38% the size of wt responses. This difference was significant over two decades of flash intensities (p-values ranging from 0.04 to 0.005; *Figure 8D*). Further descriptions of erg kinetics involved comparing the time-to-peak and rate-of-rise for the a- and b-waves. The results show that the time-to-peak for the a- and b-waves were not significantly different (*Figure 8E–F*), except for the b-wave at the highest flash intensity, which had a significantly longer time-to-peak (*Figure 8F*). To calculate rate-of-rise, the peak amplitude was divided by time-to-peak for each light flash intensity. The rate-of-rise for a-waves were nearly identical for the two genotypes (*Figure 8G*). The wt mice had a b-wave rate-of-rise that rose two- to threefold faster than the *Ribeye*-ko b-wave (p-values ranging from 0.04 to 0.009; *Figure 8H*), which mirrors the difference in b-wave amplitudes. These findings show that the ribbonless retina produced a significantly smaller b-wave under scotopic conditions; however, the kinetics were not significantly affected.

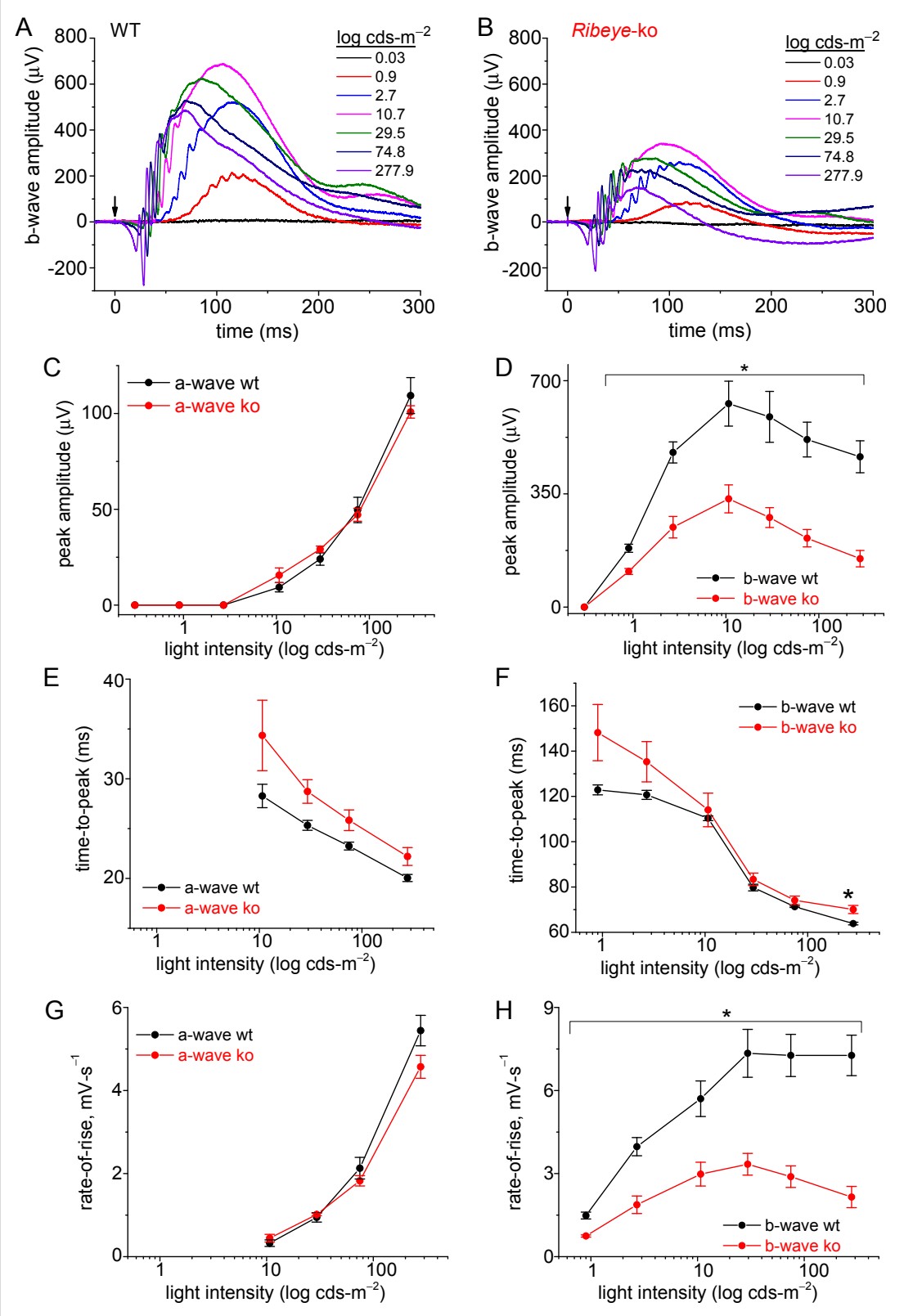

**Figure 8.** Rod-driven light responses are depressed in *Ribeye*-ko mice. (**A**) Scotopic-ergs recorded from a dark-adapted wt animal. Light flash intensities are indicated in the graph. The arrow marks the moment of the 0.1 ms light flash. Responses are presented at full band-width, without offline filtering. (**B**) Scotopic-ergs recorded from a dark-adapted *Ribeye*-ko animal. (**C**) Summary of erg a-wave amplitudes plotted over the range of light flash intensities shows no significant difference between wt and ko mice . (**D**) Summary of dark-adapted erg b-waves measured from wt and ko mice shows a

*Figure 8 continued on next page*

*Figure 8 continued*

significant difference in responses at all flash intensities (*, p<0.04 ), except at the weakest intensity tested. (**E**) The a-wave time-to-peak values were not significantly different when comparing genotypes . (**F**) The b-wave time-to-peak values were similar, except the ko responses were significantly slower at the highest flash intensity (*, p: 0.034 ). (**G**) The a-wave rate-of-rise values were not significantly different. (**H**) The b-wave rate-of-rise was significantly faster for wt than ko mice across the full range of flash intensities (*; p-values: 0.04–0.009). Average values and statistical comparisons presented in **C–H** were derived from 4 ko and 3 wt mice.

## Discussion

In this study, we set out to test whether the mammalian rod ribbon created a large RRP of SVs. This hypothesis was made decades earlier, motivated in part by results from EM studies on cat rod synapses *Rao-Mirotznik et al., 1995*; in addition, several quantitative computational studies have modeled how this synapse operates (*Rao-Mirotznik et al., 1998*; *van Rossum and Smith, 1998*; *Hasegawa et al., 2006*). What has been missing from the literature is evidence that the mammalian rod ribbon actually creates a large RRP of SVs. Therefore, the first question addressed was how many SVs were primed for release. High resolution, whole-cell $C_m$ measurements of evoked release showed that the mouse rod was able to fuse 87 SVs in a single kinetic phase with a $\tau_{depletion}$ of ~0.4 ms (*Figure 5A*). The ultrafast rate of depletion indicated the SVs were highly primed (super-primed) for fusion (*Mennerick and Matthews, 1996*; *Neher and Brose, 2018*). The significance of a single-exponential release phase is that the RRP was uniformly primed for fusion, rather than formed from a heterogeneous pool of ultrafast and fast primed states (*Grabner and Zenisek, 2013*). Furthermore, the size of the RRP of SVs was not altered when intracellular EGTA levels were changed, which is interpreted as proof that the release sites were within a few nanometers from the $Ca_v$ channels. Since the $Ca_v$1.4 channels are concentrated along the base of the ribbon, this is where the SV release sites are presumed to be located (see schematic in *Figure 9*); but, is there enough room to accommodate a RRP of 87 SVs? Confocal images of immuno-labeled mouse rod ribbons have a contour length of 1.6 µm ( *Grabner et al., 2015*), and a length of 1.7 µm has been estimated from a study using Focused Ion Beam-EM,(*Hagiwara et al., 2018*). Two separate analyses of EM tomograms estimated ~60 (*Zampighi et al., 2011*) and ~77 (*Cooper et al., 2012*) SVs were docked at the base of the ribbon. The calculated maximal number of SVs that can be packed at the base of a 1.6 µm ribbon is 86 (*Grabner et al., 2015*), which is slightly higher than experimental estimates from EM studies, but approximately the same as the estimated RRP of 87 SVs.

To test the proposal that the ribbon contributes to the formation of release sites, ribbonless rods were studied. Here, the RRP was whittled down to 22 SVs, representing a 75% reduction from wt (*Figure 7A*). This shrinkage in pool size corresponded with previously published studies that described the ribbonless rod AZ as shortened and accompanied by fewer SVs. Specifically, an EM study observed

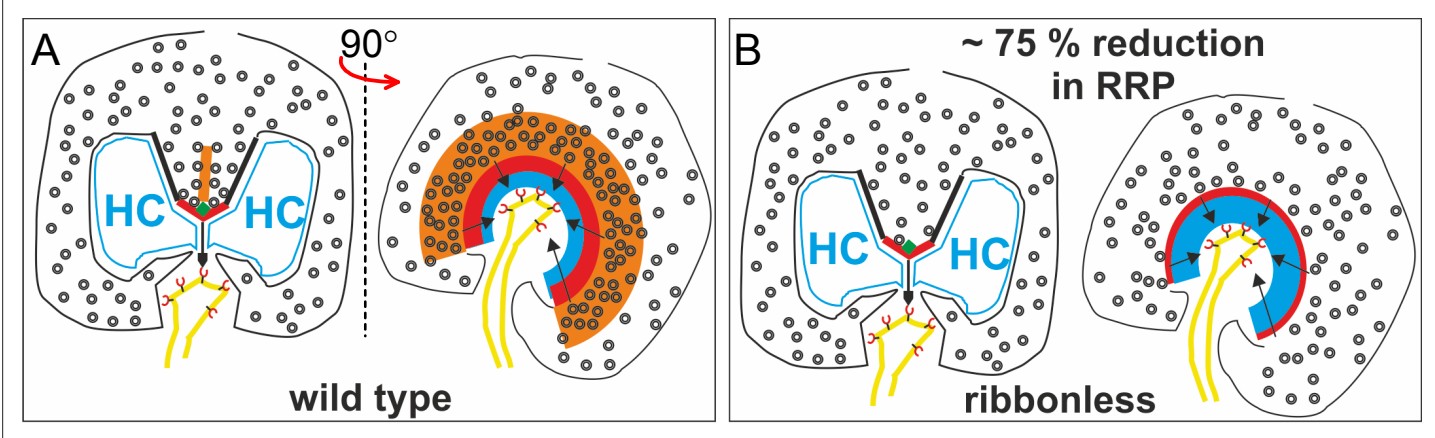

**Figure 9.** Summary cartoon of the proposed functional organization of a rod AZ. The wt AZ in (**A**) docks more SVs than the ribbonless AZ (**B**) .The two images in each panel are rotated by 90° relative to the plane of the ribbon. Legend: ribbon (orange), active zone (thick red line), arciform density (green diamond), ribbon flanked by synaptic ridges (thick black lines), horizontal cells (HC, in blue), and rod bipolar cell dendrite (yellow) with mGluR6 receptors (red). SV, synaptic vesicle.

that the density of docked SVs was reduced by 60% (for definition of docking see *Maxeiner et al., 2016*), and the rod AZ identified with Ca$_v$1.4/RIM2 immuno-staining was 50% shorter in length and rounded in the absence of the ribbon (*Maxeiner et al., 2016*; *Dembla et al., 2020*). Interestingly, the altered Ca$_v$1.4 channel staining pattern, and absence of I$_{Ca}$ facilitation in ribbonless rods (discussed below), did not alter the tight coupling of the SVs with Ca$_v$ channels. Overall a close correspondence between the size of the RRP of SVs (our study) and the number of SVs docked at the AZ (published work) was observed for wt rods. The same can be said for ribbonless rods, but the RRP and number of docked SVs were scaled down significantly (see summary illustration in *Figure 9*).

Since the overall anatomical organization of the *Ribeye*-ko retina was reported to be normal (*Maxeiner et al., 2016*; *Okawa et al., 2019*), and the scotopic erg a-waves were unaltered (*Figure 8*), the reduction in scotopic b-wave amplitudes is arguably a result of weakened transmission from rods to rbcs (*Figure 8D*) (also see: *Fairless et al., 2020*). Given the 75% reduction in evoked ΔC$_m$ measured from ribbonless rods, the altered erg b-wave amplitudes were likely, if not entirely, caused by the deficit in presynaptic release. Comparing the results from the erg and C$_m$ measurements suggests that the impairment measured with ΔC$_m$ was greater than that derived from ergs by ~15%, which may reflect an under or over representation of the ΔC$_m$ results. For instance, if in the dark wt rods released more glutamate than the postsynaptic rbc transduction pathway could encode, then the excess glutamate would not be registered in the ergs; while, a substantial release deficit may fall within the coding range of the rbc postsynaptic transduction system. Such a scenario would lead to an under representation of the deficit when probed with ergs. On the other hand, the non-physiological, strong stimulation conditions used to evoke ΔC$_m$ may have exaggerated the differences between wt and ribbonless rods.

How does ribbon loss influence retinal function under low light? Normally, synaptic glutamate concentrations at rod-rbc synapses are elevated in the dark. The consequence of a release deficit in the dark will be a reduction in synaptic glutamate, which should leave rbcs depolarized relative to wt rbcs. If true, then the ribbonless circuitry will effectively put rbcs in an 'on-state' while in the dark, which may impair the ability of the ribbonless circuitry to encode dim light stimuli (i.e., single photon responses). In the wt mouse retina, the 'primary rod pathway' handles single photon responses, and this pathway consists of the following sequence of connections (synapse type): (1) rods to rbcs (ribbon synapse), (2) rbcs to AIIs (ribbon synapse), (3) AIIs to on-cone bipolar cells (on-cbcs; gap junction), and (4) on-cbcs to on-α-ganglion cells (on-α-gcs; ribbon synapse) (for review, *Seilheimer et al., 2020*). To encode an on-response at the output layer, the rate of excitatory (glutamatergic) input to on-α-gcs is increased by light increments. When the retina is dark adapted, this pathway encodes the rate of photons captured by rods (rhodopsin isomerizations: R*) over the following range of flash intensities: 0.01– 2 R*/(rod·s); while, stronger flashes saturate the primary rod pathway (*Dunn et al., 2006*; *Ke et al., 2014*). As long as a dim background luminance is applied (<0.5 R*/(rod·s)) small increments and decrements in light can be encoded by modulating the rate of tonic excitatory inputs to on-α-gcs (*Grimes et al., 2014*). However, as background luminance >1 R*/(rod·s), the primary rod pathway rapidly adapts to light (*Dunn et al., 2006*). The adaptation is characterized by a withdrawal of tonic excitatory inputs to on-α-gcs, progressively diminishing the pathways ability to encode light decrements; thus, with light adaptation the on-α-gc responses become strongly rectified in the direction of light increments (*Grimes et al., 2014*; *Ke et al., 2014*).

On this backdrop, *Okawa et al., 2019* addressed how the ribbonless retina responded to sinusoidal-chirp light stimuli. By making whole-cell voltage-clamp recordings from on-α-gcs, they showed that the ribbonless retina encoded increments in light robustly, but light decrements were poorly encoded; which was interpreted as resulting from a reduction in tonic excitatory inputs to on-α-gcs. This basic outline is comparable to the behavior of a light-adapted primary rod pathway in as far as the responses were rectified in the direction of light increments. However, additional functional deficits indicated the ribbonless circuitry was not functioning as a normal, light adapted retina. For example, rod and cone on-pathways normally collaborate to encode contrast (*Ke et al., 2014*) and spatial frequency (*Grimes et al., 2014*) on a background luminance between 10–100 R*/(rod·s), but the ribbonless retina exhibited the greatest deficits in coding these features within this range of luminance (50 R*/(rod·s); *Okawa et al., 2019*). To assign the deficits in functional coding to the ribbonless primary rod pathway is tempting given deficits in the rod pathway outlined above; however, *Okawa et al., 2019* further showed that a normal frequency of action potentials (using on-cell patch-clamp) was generated by on-α-gcs in response to light steps from dark to 10 R*/(rod·s) for 0.5 s. This finding is consistent with

their whole-cell voltage-clamp results: robust on-responses, but it illustrates a surprising degree of functional resiliency that likely involved compensatory mechanisms. One possible explanation is that the ribbonless cone pathway, which responded normally to photopic light flashes (*Fairless et al., 2020*), helped compensate by rerouting rod signals through the secondary rod on-pathway: rods−cones−on-cbcs−on-α-gcs (*Seilheimer et al., 2020*). A step luminance of 10 R*/(rod·s) marks the point where the secondary on-pathway is normally just starting to get engaged *Ke et al., 2014*; thus, it is hard to imagine this pathway alone could compensate so robustly. Another possibility is that the ribbonless primary rod pathway utilized compensatory mechanisms in the inner retina at the rbc-AII synapse, where light adaption is normally thought to be mediated (*Dunn et al., 2006*; for review, *Demb and Singer, 2015*). Whether the primary and/or secondary rod pathway were involved will require further study. For example, by selectively deleting electrical synapses formed between rods and cones, it should be possible to study the ribbonless rod primary on-pathway more directly.

To begin to examine the biophysics of exocytosis in more detail, we first compare results from mouse rods to studies carried out on isolated Mb1 bipolars. The comparisons are rather straightforward to make since SV fusion at Mb1 synaptic terminals has been studied extensively with the whole-cell $C_m$ technique. A subpopulation of Mb1 SVs fuse with a $\tau_{ultrafast}$ ~0.5 ms (*Heidelberger et al., 1994*; *Mennerick and Matthews, 1996*; *Burrone et al., 2002*), which are rate limited by $Ca_v$ channel activation kinetics ($\tau$ ~ 0.6 ms at −10 mV, *Mennerick and Matthews, 1998*), and their release is unimpeded by elevated intracellular $Ca^{2+}$-buffering (5 mM EGTA) (*Mennerick and Matthews, 1996*). An additional population of SVs are considered to reside at greater distances from $Ca_v$ channels, because they only enter the RRP when intracellular $Ca^{2+}$-buffering is reduced (0.1 mM EGTA) (*Burrone et al., 2002*). Such heterogeneity in rates of fast release were not apparent in the recordings from mouse rods; however, we find that facilitation of $Ca_v$ channel activation kinetics occurs on a timescale that can influence the timing of ultrafast release. In response to 0.5 ms voltage steps, the RRP was depleted by 50% with 0.5 mM EGTA in the pipette, but only 4% of the RRP emptied when 10 mM EGTA was used (*Figure 5C*). The faster release onset is attributed to the acceleration of $Ca_v$ channel activation kinetics in 0.5 mM EGTA (*Figures 4F and 6A*; *Supplementary file 2*), which enhanced $Q_{Ca}$ selectively at 0.5 ms (*Figure 6E*). Interestingly, *Burrone et al., 2002* showed something very similar with respect to release, specifically a tail-current released 50% of the RRP when the Mb1 bipolars were loaded with 0.1 mM EGTA, but with higher $Ca^{2+}$-buffering (endogenous) the tail-current released only 5% of the RRP. They attributed the enhanced rate of release to the expansion of the $Ca^{2+}$-domain, but differences in $I_{Ca}$ were not reported. For comparison to mammalian bipolar cells, mouse rbcs also support ultrafast release (*Oltedal and Hartveit, 2010*), and this is mediated through tight, nanodomain coupling to $Ca_v$ channels to SVs (*Jarsky et al., 2010*).

Auditory hair cells also support a robust form of facilitation when they are pre-conditioned to elevated concentrations of free, intracellular-$Ca^{2+}$, which was achieved by pre-depolarizing the cells (*Goutman and Glowatzki, 2011*) and/or lowering intracellular $Ca^{2+}$-buffering (*Cho et al., 2011*). Release facilitation was characterized by shorter onset latencies and higher release synchrony (*Cho et al., 2011*; *Goutman and Glowatzki, 2011*; *Chen and von Gersdorff, 2019*). Notably, rat IHCs showed facilitation of $I_{Ca}$ onset, but without a change in steady-$I_{Ca}$ amplitude (*Goutman and Glowatzki, 2011*); similar to what we report for rods. However, frog HCs did not exhibit a change in $I_{Ca}$ onset (*Cho et al., 2011*). These studies show that mammalian IHCs (*Goutman and Glowatzki, 2011*) and rods accelerate $Ca_v$ channel opening (this study), and this in turn expedited release. In contrast, frog HCs (*Cho et al., 2011*) and Mb1 bipolars (*Burrone et al., 2002*) accelerate release through a distinct process that may involve a $Ca^{2+}$-dependent priming step or differences in the spatial coupling between $Ca_v$ channels and SVs, respectively.

In this study, we also found that wt $I_{Ca}$ decayed by a third with a $\tau$ of 19 ms, suggestive of $Ca_v$ channel inactivation. This was observed with 10 mM EGTA in the pipette, which is sufficient to block $Ca^{2+}$-dependent inactivation (CDI) in salamander rods (*Corey et al., 1984*; *Rabl and Thoreson, 2002*); however, $Ca^{2+}$ will need to be substituted with $Ba^{2+}$ to more definitively demonstrate the process is CDI. Voltage-dependent inactivation (VDI) may be involved, but this was not observed in salamander rods when 10 mM EGTA was used (*Corey et al., 1984*; *Rabl and Thoreson, 2002*); furthermore, VDI was not apparent in recordings from porcine (*Cia et al., 2005*) or ground squirrel (*Li et al., 2010*) rods when experiments were performed with 2 mM $Ba^{2+}$ or 10 mM BAPTA. Though we do not have a definitive answer from our recordings on rods, both CDI and VDI have been reported in studies

carried out on human Ca$_v$1.4 channels heterologously expressed in HEK cells. The molecular details are being worked out for CDI (*Haeseleer et al., 2016*; *Sang et al., 2016*); whereas, VDI has only been described as highly temperature-dependent. For instance, raising the recording temperature from 23 to 37°C increased peak-I$_{Ca}$ amplitude by 3-fold and accelerate VDI by 50-fold (*Peloquin et al., 2008*, with 20 mM BaCl2). Future studies will need to assess if VDI and/or CDI are involved.

Both of the Ca$_v$ channel gating phenomena observed in wt rods were eliminated in ribbonless rods. I$_{Ca}$ activation kinetics measured with 10 mM EGTA in the pipette showed slower activation kinetics in wt than ribbonless rods (*Figure 7D* and *Supplementary file 2*), and I$_{Ca}$ inactivation was more apparent in wt rods (*Figures 3F and 7F*). Given the steady-I$_{Ca}$ amplitude measured from wt and ko rods were similar, there were arguably comparable numbers of open channels at ~30 ms; however, ribbonless rods appeared to lack a transient facilitation in I$_{Ca}$. To better set the context, it is important to note that normal levels of Ca$_v$1.4 (α1F-subunit) protein were reported to be expressed in the retina of *Ribeye*-ko mice (*Maxeiner et al., 2016*); nonetheless, immuno-fluorescence staining for Ca$_v$1.4 (α1F) and RIM2α revealed the ribbonless AZ was 50% shorter (*Maxeiner et al., 2016*; *Dembla al., 2020*). This indicates that ribeye directly influenced the structure of the rod AZ *Maxeiner et al., 2016*, and either directly or indirectly facilitated the opening of Ca$_v$1.4 channels (this study; and see *Jean et al., 2018*). The ribbons stabilization of RIM2α and its higher molecular weight splice variants is of interest, because deletion of long forms of RIM1/2α selectively from rods significantly reduced I$_{Ca}$, reduced the frequency of miniature glutamate transporter events, and reduced the evoked ΔC$_m$, but this did not alter Ca$_v$1.4 (α1F) channel or ribeye staining patterns (*Grabner et al., 2015*). In a separate study of RIM2α-ko mice, OPL organization and rod ribbon structure were normal, but the rod pathway exhibited impaired scotopic light responses based on ergs, and excitatory inputs to horizontal cells (*Löhner et al., 2017*). From these studies, perturbation of RIM2α impaired synaptic function (i.e., Ca$_v$ channel function) without an essential role in shaping rod ribbon size or OPL organization; therefore, loss of ribeye may impair the ability of RIM2α and its splice vatiants to positively impact synaptic function at rods. Whether ribeye influences other AZ scaffolding proteins has not been thoroughly addressed, but it seems unlikely that loss of ribeye greatly impacted CAST/ELKs. When deleted individually as was done in the CAST-ko (*tom Dieck et al., 2012*), or simultaneously in the CAST/ELKs-dko (*Hagiwara et al., 2018*), the following deficits were reported: dramatic degeneration of the OPL, diminished photopic- and scotopic-erg responses, shortening of rod ribbon length, and a near elimination of rod I$_{Ca}$ (*Hagiwara et al., 2018*). A dramatic structural and functional phenotype is also observed in bassoon-ko mice. Bassoon deletion significantly impacted the anchoring of the ribbon to the AZ (*Dick et al., 2003*). Unlike ribeye and RIM2α, CAST/ELKs and bassoon are essential for normal development and maintenance of synapses in the OPL.

Our study provides new insight into the biophysics of SV fusion at the mammalian rod ribbon synapse. The results demonstrate that the rod ribbon creates multiple release sites with similar release probability in response to strong stimulations (summarized in *Figure 9*). This is driven by Ca$_v$ channels that activate at ultrafast rates. The coupling between Ca$_v$ channels and SVs is on a nano-scale, with no sign of heterogeneity in spatial coupling. Instead, release heterogeneity arose from alterations in Ca$_v$ channel facilitation, which was specific to the timing of release onset. These features were dependent on the synaptic ribbon, as ribbonless rods lacked Ca$_v$ channel facilitation and the RRP was greatly scaled down. Future studies will need to determine the stoichiometry of a release site, starting with how many Ca$_v$ channels open to trigger an SV(s) to fuse and what mechanisms impact release probability (i.e., PKA). Better insight into these matters will further our understanding of how rods convert depolarizations into synaptic signals, and ultimately how the mammalian rod pathway helps encode object motion and position.

# Materials and methods
## Animal handling
Animals were handled in accord with institutional and German national animal care guidelines. The *Ribeye* knockout (*Ribeye*-ko) mice that were first described by *Maxeiner et al., 2016*, were a kind gift from Frank Schmitz and Stefan Maxeiner (University of Saarland). The *Ribeye*-ko mice have null mutations in both alleles of the *Ribeye* gene, and were maintained on a C57BL6/J mouse background. Heterozygous males and females (*Ribeye*$^{-/+}$) were bred for experiments. Wild-type and *Ribeye*-ko

offspring littermates (male or female), between 3 and 6 months of age, were used for experiments during the daylight phase of the day/night cycle.

## Electrophysiology

Retinae were dissected at an ambient temperature of 18–20°C and then submersed in mouse extracellular solution (MES) with a low $Ca^{2+}$ concentration that had the following composition (in mM): 135 NaCl, 2.5 KCl, 0.5 $CaCl_2$, 1 $MgCl_2$, 10 glucose, 15 HEPES, and pH adjusted to 7.35 with NaOH and an osmolarity of 295 mOsm. Dissected portions of retina were absorbed onto pieces of nitrocellulose membrane mounted onto glass with the vitreal side of the retina contacting the membrane. The sclera and pigment epithelium were removed from the exposed surface of retina and then ~200 µm thick slices were made with a custom built tissue chopper. Immediately after slicing, the retinal sections (attached to the nitrocellulose membrane) were transferred to the recording chamber and arranged to be viewed in vertical cross section to optimize resolution of the OPL. Slices were washed continually with low $Ca^{2+}$ MES for approximately 5 min as they equilibrated to an ambient temperature of 30–32 °C, and then the $Ca^{2+}$ was increased to 2 mM.

The intra- and extra-cellular recording solutions have been described previously (**Grabner et al., 2015**; **Grabner et al., 2016**), and a few modifications as noted here were made to improve the $I_{Ca}$ measurements. To further block a delayed rectifier, outward $K^+$-current (**Cia et al., 2005**), the concentration of TEA was increased to 20 mM in the intracellular and 35 mM in the extracellular solutions. As previously, $Cs^+$ replaced intracellular $K^+$. Next the glutamate transporter $Cl^-$-current was previously blocked with a high concentration of DL-TBOA, 350 µM (**Grabner et al., 2016**), which is a non-selective EEAT blocker. In the current study, we used TFB-TBOA (TOCRIS), which is a high affinity blocker for EEAT1-3, at a concentration of 3 µM. It showed better stability over time, and far greater potency than the DL-TBOA (**Grabner et al., 2016**). The terminals have an $I_h$ current (**Hagiwara et al., 2018**) that was blocked by adding 5 mM CsCl to the extracellular solution (**Bader et al., 1982**). Finally, as a precaution, extracellular HEPES was elevated to 15 mM to block inhibitory proton feedback onto $Ca_v$ channels (**DeVries, 2001**). In the end, the extracellular recording solution had the following reagents (mM): 105 NaCl, 2.5 KCl, 35 TEA-Cl, 5 CsCl, 2 $CaCl_2$, 1 $MgCl_2$, 0.003 TFB-TBOA, 15 HEPES, and pH adjusted to 7.35 with NaOH, and a final osmolarity between 290 and 295 mOsm. The intracellular solution with 10 mM EGTA consisted of the following reagents (mM): 105 $CsCH_3SO_4$, 20 TEA-Cl, 1 $MgCl_2$, 5 MgATP, 0.2 NaGTP, 10 HEPES, 10 EGTA, and pH adjusted to 7.30 with CsOH to an osmolarity of 285–290 mOsm. To balance the osmolarity when EGTA was lowered to 0.5 mM, $CsCH_3SO_4$ was raised to 112 mM. The calculated liquid junction potentials ($E_{lj}$) created between the extracellular recording solution and pipette solution were: 8.9 and 9.6 mV, for 10 and 0.5 mM EGTA, respectively (**Neher, 1992**). The voltage-clamp data presented in the Results section has not been corrected for $E_{lj}$; thus, the actual applied voltages are shift by ~ −9 to −10 mV from what is stated in the manuscript.

Whole-cell patch-clamp measurements were made with a HEKA EPC-10 amplifier equipped with Patchmaster software (Lambrecht, Pfalz, Germany). The 'sine+ dc' lock-in operation mode was used to monitor changes in membrane capacitance, conductance, and series resistance. Whole-cell electrodes were fabricated from thick-wall glass capillary tubes, and their tip region was coated with Sylgard (Dow Corning). Pipette resistance was 9–11 MOhms. The cell's voltage was held at −70 mV, to which a 2 kHz sine wave with a 50 mV peak-to-peak amplitude (−95 to −45 mV) was applied. A higher sine wave frequency was used here than in a previous study on mouse rods (**Grabner et al., 2015**), because here only the axonless, soma-ribbon terminals were patched. The lock-in outputs were sampled at 20 kHz and filtered online with the low-pass $f_c$ set to 2.9 kHz. The I-$V_{step}$ protocols were sampled at 50 kHz and filtered online with the low-pass $f_c$ set to 10 kHz.

Patch-clamp recordings in this study targeted rod soma in the OPL, which contain the ribbon in the soma compartment, referred to as the 'soma-ribbon' configuration (**Hagiwara et al., 2018**). Immediately before making the on-cell seal, the extracellular solution was exchanged to an MES with 2.0 mM $Ca^{2+}$ and TEA/Cs/TBOA. After gaining whole-cell access, rods were held at a $V_m$ of −70 mV. The cells were infused for 30–40 s before the evoked release protocols began, which entailed a sequence of 5 or 7 depolarizations with stimulations given at 8 s intervals. The stimulations used to evoke release were given in the order of shortest to longest duration steps: 0.5–30 ms, and always stepping from a $V_m$ of −70 mV (rest) to −18 mV (see the Results section for a detailed explanation of the stimulation protocol used to study $\Delta C_m$ as a function of step duration.). Evoked release was studied within the

first 2 min, and then I-V protocols were performed afterwards. The passive electrical properties of the soma-ribbon measured in voltage-clamp were on average as follows, $R_{series}$: 29.7±0.6 MΩ and whole-cell capacitance $C_m$: 1.02±0.03 pF; yielding a membrane time constant $\tau$ RC ~ 30 µs (see *Figure 1C* and *Supplementary file 1*). Recordings were made at an ambient temperature of 30–32°C. Almost all recordings were made within 30–45 mins after slicing, and typically 1–2 successful recordings (cells) per mouse.

## Data analysis

The evoked $\Delta C_m$ was assessed as outlined in *Figure 2E*. Segments of the $C_m$ trace, 50 ms in length, before and after depolarization were averaged and the difference equaled the evoked $\Delta C_m$; $\Delta G_m$ and $\Delta G_s$ were calculated in the same way. Since $G_m$ is influenced by $Cl^-$-currents arising from $Ca^{2+}$-activated TMEM16A/B channels and the glutamate transporter, the following precautions were taken. For experiments with 0.5 mM EGTA in the intracellular solution, the post-depolarization segment was averaged after $I_{Cl(Ca)}$ relaxed, 75 ms after the end of the stimulation. Next, TFB-TBOA was used to block the glutamate transporter tail-currents (concentration described above). Finally, lock-in amplifier outputs were monitored between depolarization episodes, and this was done by taking 100 ms sine wave sweeps (*Figure 2A*), and the difference between 10 ms windows averaged at the start (time point: 5–15 ms) and end (t: 85–95 ms) of the 100 ms sweeps were treated as baseline/between stimulation Δ values (see *Figure 2E*).

The $I_{Ca}$ amplitude and activation time constant were determined by fitting the onset of the inward membrane current with a single exponential. When 0.5 or 10 mM EGTA was used, fits started after a 200–300 µs delay from the start of $V_{step}$, and the fit ended approximately 3 ms later (see *Figure 3A*). The exception was when 0.5 mM EGTA was used and $V_{step}$'s were made more positive than −20 mV, which accelerated $I_{Cl(Ca)}$ onset (*Figure 4B*). This left 1–2 ms to fit peak-$I_{Ca}$ when $V_{step}$ was −10 to +30 mV. The calculated $E_{Cl}$ is −51 mV, and after adding the $E_{lj}$ (9.6 mV) the $E_{Cl}$ is −41 mV. Finally, the I-V relationships were fitted with a Boltzmann equation: $I=I_{max}*(I_{min}-I_{max})/(1+exp((V_m-V_{1/2})/k))$ from a $V_{step}$ −60 mV to the $V_{step}$ corresponding to max $I_{Ca}$ (typically −10 mV). In addition, a modified form of the Boltzmann equation: $I=G_{max}*(V_m-V_{rev})/(1+exp(-(V_m-V_{1/2})/k))$, was used to fit the entire range $V_{step}$'s. Notations are as follows: I is the peak current, $V_m$ is the membrane voltage, $V_{1/2}$ is the voltage for half activation, $V_{rev}$ is the reversal potential, $G_{max}$ is the maximal conductance, and k is the slope factor. Fitting and statistical analysis were performed with Origin software (OriginLab Corporation). All experimental values are given as mean±SE. All p-values were calculated with the unpaired Student's t-test.

## ERG recordings

Mice were dark adapted overnight. They were anesthetized with an intraperitoneal injection of ketamine (0.125 mg/g) and xylazine (2.5 µg/g), and pupils dilated with 1% atropine sulfate. An AgCl wire ring-electrode was placed on the cornea, and electrical contact was made with a NaCl saline solution, plus methylcellulose to maintain moisture. A needle reference electrode was inserted subcutaneously above the nose, and a ground electrode was inserted near the tail. A custom-designed Ganzfield illuminator with 25 white LEDs was used to deliver 0.1 ms light flashes every 5 s, which were incrementally increased in intensity from 0.0003 to 0.278 cds/$m^2$ (calibrated with a Mavolux, IPL 10530). Recorded potentials were amplified, low-pass filtered at 8 kHz, and sampled at a rate of 24 kHz. Ten responses were averaged per light intensity. The ergs were low-pass filtered using an FFT set to a corner frequency of 400 Hz or 20 Hz for measurement of a- or b-wave parameters, respectively. All analyses were performed with Origin software (OriginLab Corporation).

## Converting $\Delta C_m$ into the number of fusing vesicles

To calculate the capacitance per vesicle we relied on published data. Single SV fusion events measured from mouse IHCs have an average $C_m$: 40.2 aF and SV diameter of 36.9 nm (*Grabner and Moser, 2018*). The average SV diameter, measured in mouse rod terminals, was taken from multiple published studies and yielded a diameter of 34.6 nm (32.4 nm, *Spiwoks-Becker et al., 2001*; 37.3 nm, *Fuchs et al., 2014*; 32.5 nm, *Grabner et al., 2015*; 36 nm, *Hirano et al., 2020*). The calculated $C_m$ per mouse rod SV is 37.6 aF.

## Acknowledgements

General: The authors would like to thank Stefan Thom for excellent technical assistance with the erg recordings. The authors thank Drs. Erwin Neher and Jakob Neef for critical feedback on the manuscript. Funding: This work was supported by the Max-Planck-Society (Max-Planck-Fellowship to TM) and the following funding from the Deutsche Forschungsgemeinschaft (DFG): Leibniz Award (TM); Multiscale Bioimaging Cluster of Excellence, ExC 2067 (TM); CRC 1286, project C08 (TM).

## Additional information

### Funding

| Funder | Grant reference number | Author |
|---|---|---|
| Deutsche Forschungsgemeinschaft | Leibniz Award | Tobias Moser |
| Deutsche Forschungsgemeinschaft | ExC 2067 | Tobias Moser |
| Deutsche Forschungsgemeinschaft | CRC 1286 project C08 | Tobias Moser |
| Deutsche Forschungsgemeinschaft | Multiscale Bioimaging Cluster of Excellence ExC 2067 | Tobias Moser |

The funders had no role in study design, data collection and interpretation, or the decision to submit the work for publication.

### Author contributions

Chad Paul Grabner, Conceptualization, Data curation, Formal analysis, Investigation, Methodology, Project administration, Resources, Software, Supervision, Validation, Visualization, Writing - original draft, Writing - review and editing; Tobias Moser, Conceptualization, Funding acquisition, Project administration, Resources, Supervision, Writing - review and editing

### Author ORCIDs

Chad Paul Grabner ![ORCID] http://orcid.org/0000-0001-7885-7627
Tobias Moser ![ORCID] http://orcid.org/0000-0001-7145-0533

### Ethics

All experiments complied with national animal care guidelines and were approved by the University of Götingen Board for Animal Welfare and the Animal Welfare Office of the State of Lower Saxony (permit number: 14-1391).

### Decision letter and Author response

Decision letter https://doi.org/10.7554/eLife.63844.sa1
Author response https://doi.org/10.7554/eLife.63844.sa2

## Additional files

### Supplementary files

• Transparent reporting form

• Supplementary file 1. Whole-cell patch-clamp recording parameters.

• Supplementary file 2. Activation kinetics for $Ca^{2+}$-currents measured from wild type and *Ribeye*-ko rods filled with 0.5 or 10 mM EGTA.

• Supplementary file 3. Peak-$I_{Ca}$–Voltage relationship for wild type and *Ribeye*-ko rods filled with 0.5 or 10 mM EGTA.
 Comparison of different intracellular concentrations of EGTA (within each genotype).

• Supplementary file 4. Comparison of $Ca^{2+}$-activated $Cl^-$-tail currents measured from wild type and

*Ribeye*-ko rods filled with 0.5 mM EGTA.

## Data availability

All analytic tools are described in the Methods section, and they are commercially available. All data and statistical analyses are described throughout the manuscript and in the Methods.

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
