## [Decision Letter]

**Acceptance summary:**

This paper provides elegant biophysical measures of calcium entry and vesicle fusion in mouse rod photoreceptors. These are used to estimate the size of the ready-releasable pool of vesicles and the coupling of calcium entry to release. These are important aspects of the operation of the first synapse in vision.

**Decision letter after peer review:**

Thank you for submitting your article "The mammalian rod synaptic ribbon is essential for Ca_v_ channel facilitation and ultrafast synaptic vesicle fusion" for consideration by *eLife*. Your article has been reviewed by 3 peer reviewers, including Fred Rieke as the Reviewing Editor and Reviewer #1, and the evaluation has been overseen by Richard Aldrich as the Senior Editor. The following individuals involved in review of your submission have agreed to reveal their identity: Joshua H Singer (Reviewer #2); Wallace B Thoreson (Reviewer #3).

The reviewers have discussed the reviews with one another and the Reviewing Editor has drafted this decision to help you prepare a revised submission.

The reviewers were all impressed with the quality of the data, and recognized the technical difficulty of the experiments. Concerns focused primarily on presentation, and these fell into three areas. First, the paper is dense in places and needs to be made more accessible for a general reader. One example is that the figures are quite packed, and could be split into more figures and reduced in complexity. Second, controls for rundown need to be described more clearly. Third, the changes in the ERG in the ribeye knockouts appear much smaller than would be expected from the direct measurements of exocytosis from the rods. A discussion of this apparent discrepancy is needed. In discussion, all reviewers agree these concerns were important. These and several other points are detailed in the individual reviews below.

*Reviewer #1:*

This paper describes synaptic vesicle release from rod photoreceptors. The single-cell measurements are impressive, and provide a clear set of biophysical properties of transmitter release from rods. I have few concerns, and only a few questions/suggestions, about these experiments. I found the erg measurements and their interpretation less compelling. These and some smaller comments are below,

The reduction in the erg in the RIBEYE knockout is less than might be expected given the decrease in the RRP. Similarly, retinal ganglion cells in mice lacking RIBEYE have modest response alterations when compared to control cells (see Wang et al., 2019 – this paper is relevant here and could be discussed). The robust alterations in transmitter release and seemingly smaller changes in overall circuit function seem inconsistent with each other. I do not expect the authors to have a resolution of this issue, but a discussion of it would be helpful.

Related to the above point, the biophysical measurements provide a wealth of detailed mechanistic information about the operation of the synapses. But there is much less in the paper that can be related to the operation of the synapse at physiological voltages. This may be part of why the mechanistic and functional measurements appear in conflict. This again is not a request for this data (the associated changes in capacitance and currents would be small given the small physiological changes in voltage), but it is an issue that should be clarified and discussed. A consequence of this is that the early part of the discussion about the relation of the present results to physiological function seems much less concrete than the rest of the paper.

Some descriptions are quite technical and might not be easily interpreted by a general reader. An example of this is the description of the calcium-activated chloride current as being "fully nested" in the calcium current (first full paragraph, page 8). The entire paper could be revised with an interested but non-expert reader in mind.

*Reviewer #2:*

This submitted manuscript by Grabner and Moser presents an electrophysiological study of the mouse rod presynaptic active zone (AZ). Recordings of presynaptic Ca currents and evoked exocytosis are used to assess Ca channel-release site coupling and the dynamics of exocytosis from the readily-releasable vesicle pool (RRP) under a variety of intracellular Ca^2+^-buffering conditions in wild-type and RIBEYE-knockout animals. The results show that although the RRP size is reduced in RIBEYE-knockout animals (consistent with other studies supporting the general notion that the presynaptic ribbon is an essential component of the AZ and an organizing site for the RRP), Ca channel-release site coupling is not disrupted significantly: Ca channels remain located quite close to the releasable vesicles although they are modulated differently in the absence of the ribbon.

The strength of this study is the rigorous electrophysiological analysis-the combination of Ca current and membrane capacitance recording from rod terminals-that yields high temporal resolution recordings of molecular events underlying exocytosis. Conceptual and technical problems are absent.

My major criticism addresses the presentation of the study: personally, I find that the most interesting results are obscured by the very detailed presentation of specific measurements (e.g. time constants, etc…). Revising the writing to highlight these really surprising (to me, at least) findings would make the paper far more accessible:

1) Given that the ribbon is thought to be an organizing site for exocytosis and that Ca channels are tethered to ribbon-interacting proteins, that Ca channels remain close to release sites in the absence of the ribbon is quite a surprise and provides some significant insight-or reveals limitations-about the types of protein-protein interactions that are important for Ca channel-release site coupling.

2) The fact that the absence of the ribbon affects Ca current properties but not Ca channel localization is also really interesting and, again, sets some constraints on which proteins might be interacting directly with the ribbon.

3) The finding that Ca^2+^-activated Cl channels potentially are mislocalized or disfunctional in the absence of the ribbon perhaps speaks to the role of the ribbon as a general organizer of the presynapse rather than a specific organizer of vesicles.

I leave it to the authors to bring these points forward as they see fit, but I do believe that some revision would improve the presentation.

A few additional points are worth considering and addressing:

1) Mehta et al. showed that spontaneous EPSCs were reduced in frequency and size at the cone-cone bipolar cell synapse in hibernating ground squirrels, in which the ribbon is reduced dramatically in size (and its subcellular location is altered). This speaks to a correlation between ribbon size and location and RRP size and should be mentioned.

2) Given the detailed measurements of Ca current properties that were taken, why wasn't single channel conductance estimated from noise in the tail currents? It would be worthwhile to assess this parameter, as there appear to be significant differences between single channel conductance of Cav1.4 channels in expression systems (Doering et al. 2005) and in rod bipolar cell terminals (Jarsky et al. 2010), which likely express Cav1.4.

3) I'm not sure that I like the rundown correction: doesn't it assume linearity? Is there any evidence for this?

4) The points in the discussion about RRP dynamics in fish retinal Mb1 cells and inner hair cells are well taken-there are many synapses, including these, at which the dynamics of release are quite rapid-though Mb1 cells are quite a bit different from mammalian rods as well as from mammalian rod bipolars, in which the RRP and correlations between RRP measurements and anatomical counts of vesicles have been performed (Singer and Diamond 2006; Zhou et al. 2006). The authors either should expand this section to make it more comprehensive or pare it down to remove extraneous information.

*Reviewer #3:*

This is a carefully done study describing key properties of vesicle release from mouse rods. The study is well-written and describes a series of very technically challenging experiments involving membrane capacitance measurements of vesicle fusion in mouse rods. The responses are exceedingly small, averaging only 4 fF when strongly stimulated. But the authors show very nice recordings and describe the measurements in great detail. The data describing the size of the readily releasable pool, capabilities of sustained release, properties of rod calcium currents, effects of calcium buffering on release, and changes in release observed in Ribeye KO mice that lack ribbons are all important information for understanding release at this key first synapse in the rod pathway for vision.

I recommend a greater discussion of the potential impact of rundown and depletion on their measurements. There is a mention of rundown in the calculations of sustained release in supplemental Figure 1, but it is clear from the fact that recordings were confined to the first two minutes of recording that this was a significant issue. Also, the interpulse intervals of 8 s may not be enough time for the entire releasable pool to be replenished, based on measurements in other species (e.g., Innocenti and Heidelberger, 2008; Van Hook et al., 2014). Were measurements inter-leaved to limit order effects? Were pool size measurements also adjusted for rundown?

[Editors' note: further revisions were suggested prior to acceptance, as described below.]

Thank you for resubmitting your work entitled "The mammalian rod synaptic ribbon is essential for Ca_v_ channel facilitation and ultrafast synaptic vesicle fusion" for further consideration by *eLife*. Your revised article has been evaluated by Richard Aldrich as the Senior Editor, and a Reviewing Editor.

This is a revision of a paper studying exocytosis from mouse rod photoreceptors. The paper provides elegant biophysical measures of calcium entry and vesicle fusion in wild-type and ribeye knockout mice. These are used to estimate the size of the ready-releasable pool of vesicles and the coupling of calcium entry to release. The paper has improved in revision, and is now much easier to follow. I have a few remaining suggestions for clarity:

I personally find it interesting that the functional deficits in the ribeye knockout mice are smaller than those that might be expected from the direct measurements of synaptic properties. For example, in Okawa et al. (2019), steps from darkness elicited near-identical spike responses in On-α RGCs. As you point out on the paper, the substantial alterations in operation of both rod and rod bipolar synapses might predict a strong alteration in RGC responses. The similarity of the responses to steps from darkness is not immediately consistent with the argument that you give in the discussion based on adaptation. It seems instead (or in addition) possible that there is compensation occurring. If you agree, you could add this to the discussion.

Lines 55-56: this sentence is confusing because it relates the peripheral retina and the outer retina – can you just say "the peripheral retina covers 95% of the total retinal area." Related – this paragraph treats different mammalian retinas similarly, but the distinction between central and peripheral is not very clear in mice, but obviously a big deal in primates. You might make that clearer in this paragraph.

Figure 1: the stimulus trace in Figure 1A (and in Figure 3) appears incorrect as the calcium currents recover before the stimulus ends.

Lines 130-131: I think here you mean that the Cm traces were similar across depolarizations. In any case, I would clarify what was similar.

Figure 2G: Can't you adjust the right axis scale so that you only need to plot one set of points, and can then read corresponding Cm (left axis) or SV count (right axis)? This would reinforce how you convert Cm to SVs.

Section starting on Line 139: a transition sentence stating the CaV1.4 is the calcium channel controlling release would be helpful here.

Line 336: I would consider deleting the reference to Figure 6 here since that figure has not yet been introduced – so referring to it forces the reader to look ahead. Or you could give the statement about the delay and then say something like "This conclusion was reinforced in the experiments illustrated in Figure 6A and B and described below."

---

## [Author Response]

The reviewers were all impressed with the quality of the data, and recognized the technical difficulty of the experiments. Concerns focused primarily on presentation, and these fell into three areas. First, the paper is dense in places and needs to be made more accessible for a general reader. One example is that the figures are quite packed, and could be split into more figures and reduced in complexity. Second, controls for rundown need to be described more clearly. Third, the changes in the ERG in the ribeye knockouts appear much smaller than would be expected from the direct measurements of exocytosis from the rods. A discussion of this apparent discrepancy is needed. In discussion, all reviewers agree these concerns were important. These and several other points are detailed in the individual reviews below.

Point One was addressed by making the manuscript less dense. This included splitting each Figure into multiple figures, and the description of the Results has been made accessible to the non-specialist. The ordering of Results and Figures is similar to the original submission, which should help the Reviewers evaluate the revised manuscript. The Supplemental Tables (now referred to as Supplementary Files) are unchanged.

We addressed Point Two by integrating the subject into the Results section. The revised presentation gives the general reader a better appreciation for the method and its relationship to the Results. Point Three provided us with an excellent chance to insert the current work into a systems level discussion, and highlight what is known about synaptic ribbon physiology.

Reviewer #1:This paper describes synaptic vesicle release from rod photoreceptors. The single-cell measurements are impressive, and provide a clear set of biophysical properties of transmitter release from rods. I have few concerns, and only a few questions/suggestions, about these experiments. I found the erg measurements and their interpretation less compelling. These and some smaller comments are below,The reduction in the erg in the RIBEYE knockout is less than might be expected given the decrease in the RRP. Similarly, retinal ganglion cells in mice lacking RIBEYE have modest response alterations when compared to control cells (see Wang et al., 2019 – this paper is relevant here and could be discussed). The robust alterations in transmitter release and seemingly smaller changes in overall circuit function seem inconsistent with each other. I do not expect the authors to have a resolution of this issue, but a discussion of it would be helpful.Related to the above point, the biophysical measurements provide a wealth of detailed mechanistic information about the operation of the synapses. But there is much less in the paper that can be related to the operation of the synapse at physiological voltages. This may be part of why the mechanistic and functional measurements appear in conflict. This again is not a request for this data (the associated changes in capacitance and currents would be small given the small physiological changes in voltage), but it is an issue that should be clarified and discussed. A consequence of this is that the early part of the discussion about the relation of the present results to physiological function seems much less concrete than the rest of the paper.

In the revised manuscript, we have addressed these points carefully in the Discussion, and throughout the Results section.

In the original submission, when describing wild type results we focused on the strong correlation between the size of the docked pool of SVs as judged from EM studies (previously published) and the size of the RRP of SVs as measured with dCm (our study). We made the same comparisons for the ribeye-ko, which we characterized as showing a 75 % reduction in the RRP, 60 % reduction in SV density (published work), and 60 % reduction in erg b-wave amplitude. However, we did not broaden the Discussion to consider how our study related to retinal circuitry.

In the revised manuscript, we highlight that both rod-to-rbc and rbc-to-AII signaling are highly dependent on ribeye (at least when probed with strong stimulations, and ergs), and then we propose how this might influence signaling to the ganglion cell layer. Importantly we discuss how our study may help advance the interpretation of the work by Okawa et al. 2019 (Nat Comm; Results section), who have studied the Ribeye-ko animals at the systems level. The following excerpts from that study highlight (1) how Ribeye-ko impacted signaling to the output layer of the retina, and (2) a need for more studies like ours which probe the influence of ribeye per synapse:

(1) ''the larger increment/decrement ratio in the absence of ribbons was due to smaller decrement responses. This in turn indicates less tonic excitatory synaptic input to RGCs, likely due to a lower rate of spontaneous glutamate release from bipolar cells…

(2).… These functional analyses indicate that, while RGCs continue to respond robustly in the absence of ribbons, their responses are significantly altered. More directed experiments are needed to interpret these differences mechanistically since ribbon synapses are normally present in both the photoreceptors and bipolar cells (see Discussion).''

Some descriptions are quite technical and might not be easily interpreted by a general reader. An example of this is the description of the calcium-activated chloride current as being "fully nested" in the calcium current (first full paragraph, page 8). The entire paper could be revised with an interested but non-expert reader in mind.

Fixed.

Reviewer #2:This submitted manuscript by Grabner and Moser presents an electrophysiological study of the mouse rod presynaptic active zone (AZ). Recordings of presynaptic Ca currents and evoked exocytosis are used to assess Ca channel-release site coupling and the dynamics of exocytosis from the readily-releasable vesicle pool (RRP) under a variety of intracellular Ca^2+^-buffering conditions in wild-type and RIBEYE-knockout animals. The results show that although the RRP size is reduced in RIBEYE-knockout animals (consistent with other studies supporting the general notion that the presynaptic ribbon is an essential component of the AZ and an organizing site for the RRP), Ca channel-release site coupling is not disrupted significantly: Ca channels remain located quite close to the releasable vesicles although they are modulated differently in the absence of the ribbon.The strength of this study is the rigorous electrophysiological analysis-the combination of Ca current and membrane capacitance recording from rod terminals-that yields high temporal resolution recordings of molecular events underlying exocytosis. Conceptual and technical problems are absent.My major criticism addresses the presentation of the study: personally, I find that the most interesting results are obscured by the very detailed presentation of specific measurements (e.g. time constants, etc…). Revising the writing to highlight these really surprising (to me, at least) findings would make the paper far more accessible:

Fixed.

1) Given that the ribbon is thought to be an organizing site for exocytosis and that Ca channels are tethered to ribbon-interacting proteins, that Ca channels remain close to release sites in the absence of the ribbon is quite a surprise and provides some significant insight-or reveals limitations-about the types of protein-protein interactions that are important for Ca channel-release site coupling.

Agreed.

2) The fact that the absence of the ribbon affects Ca current properties but not Ca channel localization is also really interesting and, again, sets some constraints on which proteins might be interacting directly with the ribbon.

Agreed.

3) The finding that Ca^2+^-activated Cl channels potentially are mislocalized or disfunctional in the absence of the ribbon perhaps speaks to the role of the ribbon as a general organizer of the presynapse rather than a specific organizer of vesicles.

Agreed.

I leave it to the authors to bring these points forward as they see fit, but I do believe that some revision would improve the presentation.A few additional points are worth considering and addressing:1) Mehta et al. showed that spontaneous EPSCs were reduced in frequency and size at the cone-cone bipolar cell synapse in hibernating ground squirrels, in which the ribbon is reduced dramatically in size (and its subcellular location is altered). This speaks to a correlation between ribbon size and location and RRP size and should be mentioned.

Agreed. Added to the Discussion.

2) Given the detailed measurements of Ca current properties that were taken, why wasn't single channel conductance estimated from noise in the tail currents? It would be worthwhile to assess this parameter, as there appear to be significant differences between single channel conductance of Cav1.4 channels in expression systems (Doering et al. 2005) and in rod bipolar cell terminals (Jarsky et al. 2010), which likely express Cav1.4.

In general, Jarsky 2010 is discussed as much as possible in the revised manuscript. In addition, other rbc-AII studies are highlighted to make the point that mammalian rbcs support ultrafast release and this is mediated by nano-domain coupling.

We agree that there are significant differences in the literature related to single channel conductance of L-type Cavs. This is entirely why we avoided the topic.

(3) I'm not sure that I like the rundown correction: doesn't it assume linearity? Is there any evidence for this?

Your comment is related to a portion of the manuscript that has been removed. Specifically, the dCm responses evoked with 200 ms steps were removed (originally this was presented in Results and Suppl). That data was originally presented with a correction for rundown, which was only utilized for the 200 ms stimulations since they were given after and outside the sequence of short duration steps. However, there were actually two assumptions of linearity, (1) a 'linear release rate': dCm/200 ms, and (2) as you pointed out: a linear rundown. We did not notice the 2d assumption until you pointed it out. However, the original intent of deriving a cursory estimate of sustained release (that subsequent to depletion of the RRP) would not be properly served by the 200 ms stimulations.

(4) The points in the discussion about RRP dynamics in fish retinal Mb1 cells and inner hair cells are well taken-there are many synapses, including these, at which the dynamics of release are quite rapid-though Mb1 cells are quite a bit different from mammalian rods as well as from mammalian rod bipolars, in which the RRP and correlations between RRP measurements and anatomical counts of vesicles have been performed (Singer and Diamond 2006; Zhou et al. 2006). The authors either should expand this section to make it more comprehensive or pare it down to remove extraneous information.

As noted above for (2), these works are cited in Discussion and Results.

Reviewer #3:This is a carefully done study describing key properties of vesicle release from mouse rods. The study is well-written and describes a series of very technically challenging experiments involving membrane capacitance measurements of vesicle fusion in mouse rods. The responses are exceedingly small, averaging only 4 fF when strongly stimulated. But the authors show very nice recordings and describe the measurements in great detail. The data describing the size of the readily releasable pool, capabilities of sustained release, properties of rod calcium currents, effects of calcium buffering on release, and changes in release observed in Ribeye KO mice that lack ribbons are all important information for understanding release at this key first synapse in the rod pathway for vision.I recommend a greater discussion of the potential impact of rundown and depletion on their measurements. There is a mention of rundown in the calculations of sustained release in supplemental Figure 1, but it is clear from the fact that recordings were confined to the first two minutes of recording that this was a significant issue. Also, the interpulse intervals of 8 s may not be enough time for the entire releasable pool to be replenished, based on measurements in other species (e.g., Innocenti and Heidelberger, 2008; Van Hook et al., 2014). Were measurements inter-leaved to limit order effects? Were pool size measurements also adjusted for rundown?

In short, the response to the short duration step depolarization protocol are presented the same way they were in the original manuscript, as raw values.

As for sustained release, which used a correction for rundown in the original manuscript, this data was removed (also see response to reviewer 2), because ultimately the assumption that release is linear (after depletion of the RRP) is probably unlikely based on the literature. There are many examples from salamander PRs and goldfish bipolars that show multiple, kinetic release phases following depletion of the RRP. Therefore, a cursory estimate of sustained release as originally presented in the first submission may not be helpful.

Also, the interpulse intervals of 8 s may not be enough time for the entire releasable pool to be replenished, based on measurements in other species (e.g., Innocenti and Heidelberger, 2008; Van Hook et al., 2014).

Our preliminary experiments on the axonless rods indicated the mouse rods recovered from brief stimulations (≤ 3 ms) after 1 to 2 s, and the responses reached a near saturating amplitude by 3 ms. Furthermore, previously published studies that implemented paired-pulse protocols showed that the RRP refilled in ~1 s.

1. Published dCm measurements from mammalian ground squirrel cones indicates the RRP of SVs recovered with a tau of ~0.7 s (Grabner.…DeVries 2016 Neuron). The stimulation protocols and solutions used in this study were informed by that work.

2. Based on Cm measurements, the RRP of SVs formed in salamander rods recover in ~1 sec, or less.

2a. Innocenti and Heidelberger, 2008. Whole-cell dCm was to monitor SV recovery (J Neurophysio, 2008): Abstract: This pool could be depleted with a time constant of a few hundred milliseconds and its recovery from depletion was quite rapid (tau approximately 1 s)

2b. Rabl, Cadetti, Thoreson. 2006. J Neurosci. Paired-pulse depression at photoreceptor synapses

– Abstract*:* Correlation between presynaptic and postsynaptic measures of recovery from PPD suggests that 80 –90% of the depression at these synapses is presynaptic in origin.

– Results: page 2559. Similar to the PSC, the capacitance increase evoked by the second pulse after an interstimulus interval of 100 ms was much smaller than the original response, but there was nearly complete recovery when the interstimulus interval was lengthened to 1 s.

2c. Van Hook et al., 2014 JGenPhysio van Hook….Thoreson

– Results: To explore the mechanisms underlying this effect, we recorded EPSCs in HCs evoked by stimulating cones with pairs of depolarizing steps (−70 to −10 mV, 100 ms; Figure 1).

– We found that the time course of replenishment could be fit with two exponential functions with τ_fast_ = 815 ms (76%) and τ_slow_ = 13.0 s, suggesting that the ribbon is replenished by two kinetic mechanisms, one fast and one slower (Figure 1 and Table 1).

From this study on cones: a 76 % recovery of the RRP within 0.8 sec, and the work cited above, 8 s should be enough to refill the RRP.

Future experiments can tackle this question directly by performing a paired-pulse protocol to assess recovery time of the RRP formed by mouse rods.

Were measurements inter-leaved to limit order effects?

See Results section for a justification of the stimulation protocols, and Figure 2B and 5A for results. The step depolarizations were progressively increased in duration (0.5, 1, 3, 9, 15 and 30 ms), and given in a single, forward pass.

Were pool size measurements also adjusted for rundown?

No. Raw values only.

[Editors' note: further revisions were suggested prior to acceptance, as described below.]

This is a revision of a paper studying exocytosis from mouse rod photoreceptors. The paper provides elegant biophysical measures of calcium entry and vesicle fusion in wild-type and ribeye knockout mice. These are used to estimate the size of the ready-releasable pool of vesicles and the coupling of calcium entry to release. The paper has improved in revision, and is now much easier to follow. I have a few remaining suggestions for clarity:I personally find it interesting that the functional deficits in the ribeye knockout mice are smaller than those that might be expected from the direct measurements of synaptic properties. For example, in Okawa et al. (2019), steps from darkness elicited near-identical spike responses in On-α RGCs. As you point out on the paper, the substantial alterations in operation of both rod and rod bipolar synapses might predict a strong alteration in RGC responses. The similarity of the responses to steps from darkness is not immediately consistent with the argument that you give in the discussion based on adaptation. It seems instead (or in addition) possible that there is compensation occurring. If you agree, you could add this to the discussion.

Below, we put this request in the context of previous comments from the Reviewers, and we note our attempts to better develop the part of the Discussion related to the relevance of ribbons to functional retinal circuitry.

Reviewer 1. First response letter:The reduction in the erg in the RIBEYE knockout is less than might be expected given the decrease in the RRP. Similarly, retinal ganglion cells in mice lacking RIBEYE have modest response alterations when compared to control cells (see Wang et al., 2019 – this paper is relevant here and could be discussed). The robust alterations in transmitter release and seemingly smaller changes in overall circuit function seem inconsistent with each other. I do not expect the authors to have a resolution of this issue, but a discussion of it would be helpful.

In the current version (rv2, lines 554-560) we have briefly discussed how the conditions used for assaying evoked release (dCm) may have overrepresented the release deficit. In addition, we suggest a scenario in which the ergs may have underestimated the magnitude of a rod release deficit. In the end, a major portion (60 to 75 %) of the transmission (erg) or release (dCm) are absent from ribbonless rods. Furthermore, release from ribbonless rbcs is estimated to be reduced by 80% (Maxeiner 2016). Together the deficits identified along the first two synapses of the rod primary pathway (rod-rbc and rbc-AII synapses) are significant, especially when considered in tandem. How do these defective connections influence the overall circuitry in response to light stimuli?

561 to 582 outline of the functional circuitry

583 to 611 for our detailed Discussion of Okawa et al. 2019.

Review of what was in the first revision, and what of this is maintained in the current rv2:

In the first revision, we discussed the piecewise deficits in the context of the study by Okawa et al. 2019. They measured functional light responses from on-α-ganglion cells in wt and Ribeye-ko mice. We focused on their whole-cell voltage-clamp (WC VC) recordings, because those measurements (in response to light chirps) represented the 'more detailed' dataset. Furthermore, we were motivated to Discuss their WC VC results because those results were most relatable to the seminal works by Grimes, Schwatz and Rieke 2014 (Neuron), and Ke et al., 2014 (Neuron). Notably, the WC VC results from Okawa et al. 2019 indicated significant alterations in the light responses measured from ko mice (see Figure 7d-g); in particular, light decrements were greatly reduced in the ko responses. We tried to explain the lack of response to light decrements as reflecting an impairment in the primary rod pathway. The reasoning is that in the wt retina, when weak light decrements are presented on a very dim background luminance, the primary rod pathway plays a major role. In contrast, when the pathway is light adapted, the decrements in light are poorly coded and the primary rod pathway is repurposed (Grimes et al. 2014 and Ke et al. 2014).

In revision 1, we also noted that on-responses (depolarizations in response to light increments) were robust at on-α-gcs. However, we did not attempt to explain why on-responses were strong and near normal in the ribbonless retina. The reasoning was as follows. First, in the wt retina as the primary rod pathway adapts to light, onresponses stay robust while responses to light decrements diminish. In other words, the light adapted wt retina given a sine-light stimuli has an output similar to a halfwave rectifier, rectified in the on-direction. Second, we propose that the rod secondary pathway becomes engaged (rod to cone transfer via gap junctions) as the background luminance increases and eventually cones are activated directly. Therefore, with more light the primary rod pathway might become mixed with other on-pathways that feed excitation into on-a-gcs. This introduction of other pathways is problematic since the primary rod pathway is what we have identified as being piecewise defective. In addition, the finding by Fairless et al. 2020 that photopic responses are normal in the ko mice, raises the possibility that the ribbonless cone pathway may allow the secondary rod pathway to function in near normal fashion as luminance increases.

Response from regarding our first revision.For example, in Okawa et al. (2019), steps from darkness elicited near-identical spike responses in On-α RGCs. As you point out on the paper, the substantial alterations in operation of both rod and rod bipolar synapses might predict a strong alteration in RGC responses. The similarity of the responses to steps from darkness is not immediately consistent with the argument that you give in the discussion based on adaptation. It seems instead (or in addition) possible that there is compensation occurring. If you agree, you could add this to the discussion.

We made a significant effort to address these issues in the revised manuscript rv2, and improve on the effort made in rv1.

We avoided discussing action potential (AP) results from Okawa 2019 in rv1 for reasons stated above: it was not as detailed as the WC VC results, not easily relatable to works by Grimes 2014 and Ke 2014, and lastly we were not sure if the rod-cone pathway could be involved in compensating when light steps to 10 R*/(sec/rod) were used (on a dark background). However, we appreciate the finding and have added this to rv2. To introduce this finding, we had to better describe at what point the different rod pathways are engaged (what luminance), and how they shape the outputs at the level of on-a-gcs. In the end, we propose that the rbc-AII synapse may have compensated (reset) in ways that allow the primary pathway to respond normally to light steps. In addition, we suggest that the ribbonless rods may shuttle signals to the rod-cone pathway via the secondary rod pathway (Ke et al., 2014).

In rv2, we clarify our proposal that the ribbonless retina behaves similar to a dark adapted retina. We draw the analogy to the point that the outputs of the ribbonless retina (to on-a-gcs) are rectified to on-responses, comparable to a light adapted wt retina. However, as noted by the Reviewer our interpretation was: not immediately consistent with the argument that you give in the discussion based on adaptation*.* This was a very helpful point. First in the context of discussing the AP results, one may have assumed we were suggesting the outer segments were also light adapted (in the dark). Second, if the ribbonless dark adapted retina had adapted its inner retina circuitry similar to a light adapted wt retina, then dim light should be less efficiently transmitted via the rbc-AII pathway; thus, less likely to produce a normal frequency of APs.

Therefore, here we addressed compensation as outlined above (resetting of the rbc-AII synapse, or rerouting rod signals via the secondary rod-cone pathway). Third, and apart from APs, we took into consideration that a light adapted wt primary rod pathway influences contrast and spatial coding between 10 to 100 R*/(sec/rod) (Ke et al. , 2014). In the paper by Okawa et al. 2019 the Ribeye-ko responses to chirps on a mean luminance of 50 R*/(sec/rod) were the most significantly impaired with respect to encoding spatial frequency and contrast (Figure 7g; Owaka). This finding is inconsistent with a light adapted primary rod pathway, but instead indicates a

functional impairment that may reflect deficits in the primary pathway; however, the

rod secondary on-pathway or cone deficits may also contribute to the impaired responses.

We did not discuss the AP-4 induced slowing of miniEPSC frequency as reported by

Okawa et al. 2019. Space prohibited us from commenting on the AP-4 data in the Discussion, and the most we would be able to say is that it indicated no obvious changes in the system's ability to signal a dark state and mepscs were unaltered at between on-cbcs and on-a-gcs.

Lines 55-56: this sentence is confusing because it relates the peripheral retina and the outer retina – can you just say "the peripheral retina covers 95% of the total retinal area." Related – this paragraph treats different mammalian retinas similarly, but the distinction between central and peripheral is not very clear in mice, but obviously a big deal in primates. You might make that clearer in this paragraph.

This has been removed.

Figure 1: the stimulus trace in Figure 1A (and in Figure 3) appears incorrect as the calcium currents recover before the stimulus ends.

The stimulus traces (schematics) have been reformatted: Figure 1D, and Figure 3F and G. Figure 1C illustrates the uncompensated capacitive transient, which decays before the step voltage is returned to the holding voltage

Lines 130-131: I think here you mean that the Cm traces were similar across depolarizations. In any case, I would clarify what was similar.Figure 2G: Can't you adjust the right axis scale so that you only need to plot one set of points, and can then read corresponding Cm (left axis) or SV count (right axis)? This would reinforce how you convert Cm to SVs.Section starting on Line 139: a transition sentence stating the CaV1.4 is the calcium channel controlling release would be helpful here.Line 336: I would consider deleting the reference to Figure 6 here since that figure has not yet been introduced – so referring to it forces the reader to look ahead. Or you could give the statement about the delay and then say something like "This conclusion was reinforced in the experiments illustrated in Figure 6A and B and described below."

Thank you! That was a helpful fix.